# Adversarial Training with Complementary Labels: On the Benefit of Gradually Informative Attacks

**Jianan Zhou**[1*]   **Jianing Zhu**[1*]   **Jingfeng Zhang**[2]   **Tongliang Liu**[3]
**Gang Niu**[2†]   **Bo Han**[1†]   **Masashi Sugiyama**[2,4]

[1]Hong Kong Baptist University    [2]RIKEN Center for Advanced Intelligence Project
[3]TML Lab, The University of Sydney    [4]The University of Tokyo

{csjnzhou, csjnzhu, bhanml}@comp.hkbu.edu.hk
jingfeng.zhang@riken.jp    tongliang.liu@sydney.edu.au
gang.niu.ml@gmail.com    sugi@k.u-tokyo.ac.jp

## Abstract

Adversarial training (AT) with imperfect supervision is significant but receives limited attention. To push AT towards more practical scenarios, we explore a brand new yet challenging setting, i.e., AT with complementary labels (CLs), which specify a class that a data sample does not belong to. However, the direct combination of AT with existing methods for CLs results in consistent failure, but not on a simple baseline of two-stage training. In this paper, we further explore the phenomenon and identify the underlying challenges of AT with CLs as intractable adversarial optimization and low-quality adversarial examples. To address the above problems, we propose a new learning strategy using gradually informative attacks, which consists of two critical components: 1) Warm-up Attack (Warm-up) gently raises the adversarial perturbation budgets to ease the adversarial optimization with CLs; 2) Pseudo-Label Attack (PLA) incorporates the progressively informative model predictions into a corrected complementary loss. Extensive experiments are conducted to demonstrate the effectiveness of our method on a range of benchmarked datasets. The code is publicly available at: https://github.com/RoyalSkye/ATCL.

## 1   Introduction

Deep neural networks are vulnerable to adversarial examples [15], which motivates the development of various defensive methods [30, 21, 24, 2] to mitigate the arisen security issue. As one of the most effective and practical defensive methods, adversarial training (AT) [24] has been widely studied [41, 5, 10]. Specifically, it formulates the problem as min-max optimization, which generates adversarial data during training and minimizes the training loss of the generated adversarial variants. In this way, the models are equipped with adversarial robustness against human-imperceptible perturbations within the neighborhood of inputs. Although AT with perfect supervision (e.g., ordinary labels) has been thoroughly studied by previous works [25, 33, 44, 21, 34, 43, 11], the more common learning scenarios with imperfect supervision [45, 42] has only received limited attention.

Complementary labels (CLs) [17], which convey partial label information by identifying a class that a data sample does not belong to, are one of those practical and promising imperfect supervision in weakly supervised learning [3, 27]. Several works [39, 18, 7, 13, 38, 14] recently focused on learning with CLs, where the model trained only with CLs through their designed losses (i.e., complementary loss $\bar{\ell}$) is expected to predict ordinary labels accurately during inference. Their success illustrates the

---

*Equal contribution.

†Corresponding authors: Bo Han (bhanml@comp.hkbu.edu.hk) and Gang Niu (gang.niu.ml@gmail.com).

36th Conference on Neural Information Processing Systems (NeurIPS 2022).

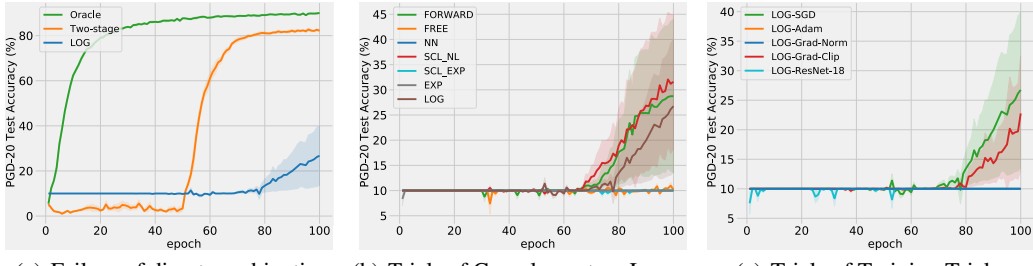

|                          |                          |                          |
|:------------------------:|:------------------------:|:------------------------:|
| (a) Failure of direct combination | (b) Trials of Complementary Losses | (c) Trials of Training Tricks |

Figure 1: (a) in contrast to a simple two-stage baseline, the direct combination of AT with CLs (LOG [13]) results in the worse performance with the *failed training curve*. The two-stage baseline consists of complementary learning in the first 50 epochs and AT with predicted labels in the last 50 epochs; (b) we try various complementary loss functions while all of them result in *consistent experimental failure* in AT with CLs; (c) several training alternatives (e.g., optimizer) are tried with no success. We report mean with standard errors for PGD20 test accuracy on Kuzushiji within multiple random trials. The training on the direct combination of AT with CLs is significantly unstable.

possibility of training ordinary classifiers even when all the labels given for training are wrong (except that the data and labels are statistically independent). Due to the low acquisition cost and privacy preservability, it has been applied to online learning [19] and medical image segmentation [31].

To push AT towards more practical scenarios, we explore a brand new yet challenging setting, i.e., *AT with CLs*. In this paper, our interest is how to equip machine learning models with adversarial robustness when all given labels in a dataset are wrong (i.e., CLs), which is of scientific interests to both research areas of weakly supervised learning and adversarial learning. To conduct AT with CLs, a straightforward attempt is to regard $\bar{\ell}$ as the training objective of the min-max optimization [15, 24].

However, the direct combination of AT with existing methods [39, 18, 7, 13] for learning with CLs results in consistent failure, but not on a simple baseline of two-stage training (as shown in Figure 1). To be specific, we mainly consider the single complementary label (SCL) with the uniform assumption [17, 18, 7], where labels other than the true label are chosen as a CL with the same probability. In complementary learning, the complementary losses $\bar{\ell}$ are obtained by certain correction techniques [17, 39], which guarantee the training with $\bar{\ell}$ on complementary datasets achieves similar performance to the training with the ordinary loss $\ell$ on ordinary datasets. In contrast, when conducting AT with CLs, most experiments (as shown in Figure 1 with different $\bar{\ell}$ and optimizers) consistently show failed training curves except for a simple two-stage baseline, which consists of a natural complementary learning phase and an AT phase using the predicted labels of the first-stage model.

The above fact motivated us to further investigate AT with CLs, and we identified its unique challenges as *intractable adversarial optimization* and *low-quality adversarial examples* (as verified in Figure 2). Specifically, we first theoretically prove that conducting AT with the complementary risk is statistically consistent to that with the ordinary risk when training with the general backward corrected loss [18]. Although the guarantee holds with sufficient data, it is not preserved any more with a limited number of CLs given in practice [7] (see Eq. (8) in Section 4.1). Empirically, we analyze the training dynamics of AT with CLs (e.g., gradient directions and norms), and observe that adversarial optimization with CLs through $\bar{\ell}$ suffers from large gradient variance, with the gradient vanishing issue at the early stage of training. Moreover, we also find that the complementary loss as the objective of adversarial optimization fails to generate high-quality adversarial examples due to the less informative nature [17] of CLs and the implicit way of attacking ordinary labels (i.e., attacking $\sum_{j \neq \bar{y}} p_\theta(j|x)$ instead).

To mitigate the problems, we propose a new learning strategy using gradually informative attacks (in Algorithm 1) for AT with CLs, which consists of two critical components, i.e., *Warm-up Attack (Warm-up)* and *Pseudo-Label Attack (PLA)*. We introduce the Warm-up to ease the intractable optimization with CLs, which is gradually transferred from natural training to AT by controlling the adversarial budget. We design the PLA to improve the low-quality adversarial generation, which incorporates the informative predictions from the progressively discriminative model itself. Overall, we propose a unified framework using a flexible scheduler to adjust two critical components during training.

Our contributions are summarized as follows. 1) Setting level: we study AT with CLs, which is a practical yet thus far unexplored setting. 2) Challenge level: we discover the substantial experimental failures in the direct combination of AT with existing methods for learning with CLs (in Figure 1), and provide the underlying insights of the observed phenomenon from both theoretical and empirical perspectives (in Sections 4.1 and 4.2). Specifically, we identify two challenges as intractable adversarial optimization and low-quality adversarial examples. 3) Methodology level: we accordingly propose a unified framework, using Warm-up and PLA with a controllable scheduler to mitigate the problems (in Sections 4.3). 4) Experimental level: we conduct extensive experiments to verify the effectiveness of our method, and to understand the benefits of two critical components (in Section 5).

## 2 Related Work

**Complementary Learning.** Learning with CLs was introduced by [17], where the SCL with the uniform assumption was firstly considered. An unbiased risk estimator (URE) was derived by modifying a specific loss function (e.g., one-versus-all or pairwise comparison) that satisfies a symmetric condition. Continuing with this work, [18] derived a general URE for arbitrary loss functions and models, and further proposed non-negative correction and gradient ascent methods to cope with the overfitting issue of URE in practice. Although URE has good statistical properties, it has inferior empirical performance due to huge gradient variance. [7] mitigated this problem by proposing a surrogate complementary loss framework, which is a biased risk estimator trading zero bias with reduced variance. Other than the uniform assumption, [39] considered the biased CLs, where labels other than the true label are chosen as a CL with different probabilities due to the annotator's bias. In addition to the SCL setting, [13] derived an URE of the ordinary risk for multiple complementary labels, and further improved it by minimizing properly chosen upper bounds. Moreover, there are also several works less related to the loss correction in complementary learning, such as introducing regularization [38] or reweighting [14] during the training process.

**Adversarial Training.** As one of the most effective defensive methods, AT has been widely studied to improve the robustness of deep learning models [1, 12, 37, 36, 26, 35, 6]. The standard AT [24] formulated the problem as min-max optimization. [4] proposed CAT to mitigate the catastrophic forgetting and the generalization issues of AT. [41] decomposed the prediction error into the natural and boundary errors, and proposed TRADES to balance the classification performance between the natural and adversarial examples. [43] resolved the trade off between robustness and accuracy via reweighting adversarial data by their geometrical information to the decision boundary. In contrast to previous works, only limited effort has been made to explore AT with imperfect supervision [27, 18]. [45] explored the interaction of AT with noisy labels (NLs), and found that AT itself is NLs correction since the number of PGD steps could be used to improve sample selection quality. [42] investigated the effects of NLs injection on the adversarial optimization, and proposed NoiLIn to mitigate the robust overfitting issue based on the empirical observation that NLs are not always detrimental.

## 3 Preliminaries

In this section, we provide preliminaries for ordinary learning, complementary learning and AT. For complementary learning, we consider the SCL setting with the uniform assumption [17, 18, 7]. For AT, we focus on the standard method proposed in [24].

### 3.1 Ordinary Learning

Let $\mathcal{X} \subset \mathbb{R}^d$ be the input space, and $\mathcal{Y} \in [K] := \{1, 2, \ldots, K\}(K > 2)$ be the label space. The data $\{(x_i, y_i)\}_{i=1}^n$ is sampled independently and identically from the joint distribution $\mathcal{D}$ with density $p(X, Y)$, which could be further decomposed into $p(X)p(Y|X)$. We define a class-probability function $\eta_i(x) = p(Y = i|X = x)$. The decision and loss functions are defined as $g : \mathcal{X} \to \mathbb{R}^K$ and $\ell : \mathcal{Y} \times \mathbb{R}^K \to \mathbb{R}^+$, respectively. The goal of ordinary learning is to train a classifier $f(x) : \mathcal{X} \to \mathcal{Y}$, which is expected to predict the correct label by $\arg\max_i g(x)_i$. The ordinary risk is defined as

$$R(g; \ell) = E_{(x,y)\sim\mathcal{D}}[\ell(y, g(x))]. \tag{1}$$

## 3.2 Complementary Learning

In complementary learning, the label space is transformed into $\bar{\mathcal{Y}}$, where each ordinary label $y$ is flipped into $\bar{y} \in \bar{\mathcal{Y}}$ with a class-dependent probability $p(\bar{y}|y)$. Then, the data is sampled from a different joint distribution $\bar{\mathcal{D}}$ with density $p(X, \bar{Y})$. We assume the transition matrix $Q \in \mathbb{R}^{K \times K}$ is invertible, where $Q_{ij} = p(\bar{Y} = j|Y = i)$. With the uniform assumption, it takes 0 on diagonals and $\frac{1}{K-1}$ on non-diagonals. The class-probability function is then written as

$$\bar{\eta}_i(x) = p(\bar{Y} = i|X = x) = \sum_{j \neq i} p(\bar{Y} = i|Y = j)p(Y = j|X = x) = \sum_{j \neq i} Q_{ji}\eta_j(x). \tag{2}$$

The complementary risk is defined as $\bar{R}(g; \bar{\ell}) = E_{(x,\bar{y}) \sim \bar{\mathcal{D}}}[\bar{\ell}(\bar{y}, g(x))]$, where $\bar{\ell}$ is a complementary loss function. Typically, it can be derived by the loss correction, which is a technique that ensures risk or classifier consistency in the weakly supervised setting [27, 17]. The backward correction is a risk consistent algorithm that ensures $\bar{R}$ is a statistically consistent risk estimator of the ordinary risk $R$, while the forward correction is a classifier consistent algorithm that guarantees the classifier learned from $\bar{\mathcal{D}}$ using $\bar{\ell}$ converges to the optimal one learned from $\mathcal{D}$ using $\ell$. In complementary learning, [17, 18, 13] focused on the backward correction that multiplies the loss by $Q^{-1}$, while [39] proposed a forward correction method via multiplying the model prediction by $Q$. Other methods (e.g., a bounded loss) can also be used to derive complementary losses [7, 13]. The detailed complementary loss functions are provided in Table 2 of Appendix B.

**General Backward Correction.** Among this line of research, a notable work [18] derived an URE of the ordinary risk, without any restriction on the used loss and model. Given the notations borrowed from [18, 7]: $\ell(g(x)) = [\ell(1, g(x)), \ell(2, g(x)), \ldots, \ell(K, g(x))]$ and $e_i$ is the one-hot column vector with one on the $i_{\text{th}}$ entry, the URE is derived as follows:

**Proposition 1.** *The ordinary risk can be expressed in terms of CLs as follows,*

$$R(g; \ell) = E_{(x,y) \sim \mathcal{D}}[\ell(y, g(x))] = E_{(x,\bar{y}) \sim \bar{\mathcal{D}}}[\bar{\ell}(\bar{y}, g(x))] = \bar{R}(g; \bar{\ell}), \tag{3}$$

*when $\bar{\ell}$ is rewritten as*

$$\bar{\ell}(\bar{y}, g(x)) = e_{\bar{y}}^T (Q^{-1})\ell(g(x)), \tag{4}$$

*With the uniform assumption, $\bar{\ell}$ can be further rewritten as*

$$\bar{\ell}(\bar{y}, g(x)) = -(K - 1)\ell(\bar{y}, g(x)) + \sum_{j=1}^{K} \ell(j, g(x)). \tag{5}$$

With this expression, we can obtain an URE of the ordinary risk only from CLs.

## 3.3 Adversarial Training

AT has been a commonly used defensive technique that equips deep learning models with adversarial robustness against imperceptible perturbations within a small neighborhood of input. Here, we consider conducting AT with ordinary labels (e.g., $\ell$ is the cross-entropy loss). Generally, it can be formulated into a min-max optimization problem:

$$\min_{\theta} E_{(x,y) \sim \mathcal{D}}[\ell(y, g(\tilde{x}))], \text{ with } \tilde{x} = \arg\max_{\tilde{x}_i \in \mathcal{B}_\epsilon[x]}[\ell(y, g(\tilde{x}_i))], \tag{6}$$

where $\mathcal{B}_\epsilon[x] = \{\tilde{x} : \|x - \tilde{x}\|_p \leq \epsilon\}$ is the closed ball centered at input $x$ with radius $\epsilon > 0$ under $l_p$-norm threat models, and $\tilde{x}$ is the adversarial data found within $\mathcal{B}_\epsilon[x]$. Empirically, the solution of inner maximization is approximated by projected gradient descent (PGD) as follows:

$$x^{(t+1)} = \Pi_{\mathcal{B}_\epsilon[x]}[x^{(t)} + \alpha\text{sign}(\nabla_{x^{(t)}}\ell(y, g(x^{(t)})))], \tag{7}$$

where $\alpha$ is the step size, $\Pi$ is the projection operator that projects the adversarial data back to the epsilon ball, $x^{(t)}$ is the adversarial data found at step $t$, and $x^{(0)}$ is initialized by natural data or natural data with a small Gaussian or uniformly random perturbation. The adversarial data is updated iteratively in the direction of loss maximization until a stop criterion is satisfied.

# 4 Adversarial Training with Complementary Labels

In this section, we first provide some theoretical insights for AT with CLs. Then, we empirically analyze the experimental failure and identify the critical challenges as *intractable adversarial optimization* and *low-quality adversarial examples* when conducting AT with corrected complementary losses. Accordingly, we propose a new learning strategy with gradually informative attacks.

## 4.1 Theoretical Analysis

Without loss of generality, below we analyze the general backward correction [18] theoretically, and other popular complementary loss functions on the SCL setting.

**Proposition 2.** *For the general backward correction, conducting AT on the complementary risk is equivalent to that on the ordinary risk. However, this is not the case on their empirical risks:*

$$\min_\theta E_{x \sim p(X)} \max_{\tilde{x} \in \mathcal{B}_\epsilon[x]} E_{y \sim p(Y|X=x)}[\ell(y, g(\tilde{x}))] = \min_\theta E_{x \sim p(X)} \max_{\tilde{x} \in \mathcal{B}_\epsilon[x]} E_{\bar{y} \sim p(\bar{Y}|X=x)}[\bar{\ell}(\bar{y}, g(\tilde{x}))],$$

$$\min_\theta \frac{1}{n} \sum_{i=1}^n \max_{\tilde{x}_i \in \mathcal{B}_\epsilon[x_i]} [\ell(y_i, g(\tilde{x}_i))] \neq \min_\theta \frac{1}{n} \sum_{i=1}^n \max_{\tilde{x}_i \in \mathcal{B}_\epsilon[x_i]} [\bar{\ell}(\bar{y}_i, g(\tilde{x}_i))]. \tag{8}$$

The detailed proof is provided in Appendix A.2. In brief, when considering the general backward corrected loss, the maximization induced by AT keeps the statistical properties of the complementary risk, and it still yields a consistent risk estimator. In theory, the adversarial optimization of the complementary risk with the general backward corrected loss is statistically consistent to that of the ordinary risk with the ordinary loss. But in practice, we can only obtain limited CLs (e.g., the sample size of $E_{\bar{y} \sim p(\bar{Y}|X=x)}$ is one on the SCL setting), which causes the inconsistency between their empirical risks (i.e., the solutions to their inner maximization are different), and hence leads to a difficult adversarial optimization (as verified in Appendix D.2). The solution to inner maximization of the complementary risk is equivalent to that of the ordinary risk if and only if maximizing the weighted loss over all candidate CLs, that is $E_{\bar{y} \sim p(\bar{Y}|X=x)}[\bar{\ell}(\bar{y}, g(\tilde{x}))]$.

From the perspective of AT, we expect to find the adversarial examples $\tilde{x}$ within the neighborhood of input $x$ that make the model predictions on the ordinary label $p_\theta(y|\tilde{x})$ as low as possible. When we have ordinary labels, it is quite straightforward to achieve this objective by maximizing the cross-entropy $\ell = -\log p_\theta(y|\tilde{x})$. However, if we further consider recent proposed methods in addition to the general backward correction in complementary learning, most of $\bar{\ell}$ mathematically try to maximize $\sum_{j \in \mathcal{Y} \setminus \{\bar{y}\}} p_\theta(j|\tilde{x})$ (or equivalently minimize $p_\theta(\bar{y}|\tilde{x})$) as shown in Table 2. Maximizing $\bar{\ell}$ leads to the minimization of $\sum_{j \in \mathcal{Y} \setminus \{\bar{y}\}} p_\theta(j|\tilde{x})$, which cannot guarantee the minimization of $p_\theta(y|\tilde{x})$. In such a case, the inner maximization may serve as a random attacker without prior knowledge. Hence, corrected complementary losses fail to achieve the objective of AT, and low-quality adversarial examples may be generated, which is detailedly discussed in the following part.

## 4.2 Empirical Analysis

We conduct the experiment on Kuzushiji to justify the previous theoretical insights. For the upper panels of Figure 2, we focus on the issue of *intractable adversarial optimization* by adversarially training the model using several loss functions separately, with three random trials. For the lower panels of Figure 2, we focus on the issue of *low-quality adversarial examples* by measuring the different adversarial examples generated by various loss functions, given the same optimization model (i.e., the *oracle* model, which is trained using AT with ordinary labels). We show the average results over different seeds and training samples, respectively. Without loss of generality, here we only show the results of one complementary loss (i.e., LOG [13]), and our proposed method (i.e., *Warm-up+PLA* in Section 4.3). The detailed experiment settings and more comprehensive empirical results with other complementary losses and datasets are provided in Appendix C.

**Intractable Adversarial Optimization.** In the outer minimization of AT, we first collect the gradient direction (with respect to the model parameters) of each data in one mini-batch (i.e., 256), and then compute the trace sum of their covariance matrix (the average result for each epoch is shown

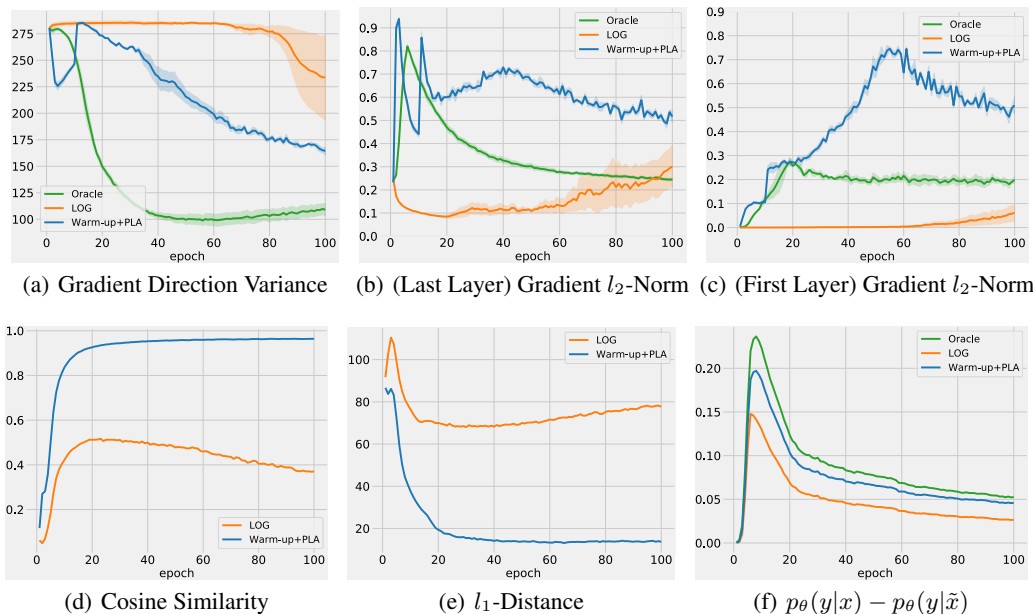

Figure 2: [*Intractable Adversarial Optimization*] (a) the trace sum of covariance matrix of gradient directions w.r.t. the parameters of the first layer; (b) the $l_2$-norm of stochastic gradients of the last layer; (c) the $l_2$-norm of stochastic gradients of the first layer; [*Low-quality Adversarial Examples*] (d) the cosine similarity of gradient directions (i.e., $\text{sign}(\nabla_x \bar{\ell}(x, \bar{y}; \theta))$) with the oracle in the inner maximization; (e) the $l_1$-distance of the constructed adversarial examples with the oracle; (f) the difference of model predictions of natural data and adversarial data on the ordinary label.

in the upper-left panel of Figure 2). A larger value reflects a group of gradient directions with a bigger variance. Note that the result of LOG nearly reaches the maximum trace sum of covariance matrix of the gradient directions, which results in its consistent failure at the early stage of training. We also compute the average $l_2$-norm of stochastic gradients of the last and first layer as shown in the upper-middle and upper-right panels of Figure 2, respectively. Although some gradient signals exist for LOG in the last layer, the gradients back-propagated to the first layer is so small that gradient vanishing problem occurs at the early stage of training. This phenomenon is similar to that observed in the standard AT, which is conjectured as the prevalence of dead neurons caused by a large radius of epsilon ball [23]. Moreover, considering AT with CLs, it is expected that the gradient norm tends to be small since the observed variance of gradient directions in one mini-batch is extremely large. The above observations indicate the adversarial optimization with CLs is difficult, and it is rational to use the method (i.e., our Warm-up+PLA) to ease the optimization.

**Low-quality Adversarial Examples.** In the inner maximization of AT, given the same optimization model, we measure the quality of generated adversarial examples by computing the cosine similarity of gradient directions (with respect to the same data) and the $l_1$-distance of constructed adversarial examples, with the oracle examples generated by ordinary label (in the lower-left and lower-middle panels of Figure 2). A larger cosine similarity and a lower $l_1$-distance indicate a stronger resemblance to the oracle, but not necessarily a stronger attack (or high-quality adversarial examples). Hence, we further compute the difference of model predictions of natural data and adversarial data on the ordinary label during training (in the lower-right panel of Figure 2). A bigger difference explicitly demonstrates a stronger attack. The results show that the complementary loss as the objective of AT with CLs fails to generate high-quality adversarial examples as our analysis in Section 4.1.

### 4.3 Methodology

To address the previous problems, we accordingly propose a new strategy using gradually informative attacks to ease the intractable adversarial optimization and improve the quality of adversarial examples, respectively. Our algorithm is concretely presented in Algorithm 1. Specifically, it consists of Warm-

**Algorithm 1** AT with CLs Using Gradually Informative Attacks

---

**Input:** number of epochs: $E$, number of batches: $B$, number of class: $K$, radius of epsilon ball: $\epsilon$, number of attack steps: $k$, step size: $\alpha$, hyperparameter for exponential moving average of cashed model predictions: $\beta$; hyperparameter for label weights: $\gamma$;

**Output:** robust model $\theta_R$;

1:   $p_c \leftarrow \frac{1}{K-1}(\mathbf{1} - e_{\bar{y}})^T$ for all data
2:   **for** e $= 1, \ldots, E$ **do**
3:      $\beta, \gamma \leftarrow$ Update hyperparameters
4:      $\epsilon_e, \alpha_e, k_e \leftarrow$ Adversarial budget scheduler                          ▷ Warm-up Attack
5:      **for** b $= 1, \ldots, B$ **do**
6:          $x, \bar{y} \leftarrow$ Sample a batch of data
7:          $p_\theta(x) \leftarrow \text{Softmax}(\theta(x)), \ p_c(x) \leftarrow \beta p_c(x) + (1-\beta)p_\theta(x), \ p_c(\bar{y}|x) \leftarrow 0$
8:          $\hat{y} \leftarrow \arg\max_{j \neq \bar{y}} p_c(j|x)$
9:          $\tilde{x} \leftarrow$ Approximate the solutions of $\max \bar{\ell}(x, \bar{y}; \theta)$ (Eq. 10) by PGD($\epsilon_e, \alpha_e, k_e$)
10:        $\mathcal{L} = -(K-1)\log(\gamma \sum_{j \neq \bar{y}} p_\theta(j|\tilde{x}) + (1-\gamma)p_\theta(\hat{y}|\tilde{x}))$        ▷ Pseudo-Label Attack
11:        $\theta \leftarrow \text{SGD}(\theta, \nabla_\theta \mathcal{L})$
12:      **end for**
13:      Evaluate($\epsilon, \alpha, k$)
14: **end for**

---

up Attack and Pseudo-Label Attack, whose dynamics are controlled by a flexible scheduler:

$$\mathbb{T}(a, e, E_s) = \begin{cases} \dfrac{a}{2} \cdot (1 - \cos(\min(\dfrac{e}{E_s}, 1) \cdot \pi)) & \text{Warm-up Attack,} \\[2ex] a \cdot (1 - \min(\dfrac{e}{E_s}, 1)) & \text{Pseudo-Label Attack,} \end{cases} \tag{9}$$

where $a$ is the maximum value of the scheduled variable (i.e., $\epsilon_{\max} = \epsilon$ in Warm-up Attack and $\gamma_{\max} = 1$ in Pseudo-Label Attack), $e$ is the epoch index, and $E_s$ is the duration of the scheduler. Both the proposed attacks can be controlled by a flexible scheduler with the increasing and decreasing trends respectively (e.g., see the example in Figure 3). Below, we describe each component in detail.

**Warm-up Attack.** Adversarial optimization is harder compared with ordinary optimization [23], and CLs further strengthen the degree of hardness as observed in our empirical analysis. The major difference of AT with CLs from the standard AT is the labels used in the adversarial generation. As shown in Figure 2, training with the adversarial examples generated by different labels results in the different gradient variance. Since the adversarial generation on CLs induces large gradient variance which causes the difficult adversarial optimization, we can adjust the adversarial generation to control the variance introduced into the optimization. To be specific, we adopt a dynamic adversarial budget to gradually enhance the degree of adversarial generation. Typically, adversarial budgets consist of the radius of epsilon ball $\epsilon$, number of attack steps $k$, and step size $\alpha$. Here, we mainly focus on the dynamics of $\epsilon$ in Eq. (9). The radius of the epsilon ball is increased from 0 to $\epsilon$ within $E_s$ epochs, and is kept constant until the end of adversarial optimization. The number of attack steps is constant ($k_e = k$), and the step size is increased proportionally to $\epsilon$ ($\alpha_e = \frac{\epsilon_e}{\epsilon}\alpha$). In this way, the gradually increased adversarial budget can ease the difficulty of adversarial optimization.

**Pseudo-Label Attack.** Due to the existence of an adversarial budget scheduler, at the early stage of optimization, simple patterns may be prevalent in adversarial data found within an extremely small epsilon ball, which is helpful for the self-evolvement of a discriminative model. The model tends to assign high confidence to the potential ordinary label, while low confidence to others. This informative model prediction is a strong supplementary information that is promising for improving the quality of the adversarial generation. Therefore, we propose to optimize the objective consisting of the convex combination outputs of partial labels (labels other than CLs) and predicted ordinary label. This transforms the whole process into an optimization that starts with learning with CLs using certain correction techniques, and gradually moves to learning with predicted labels regarding them as ordinary labels. The loss function can be defined as follows:

$$\bar{\ell}(x, \bar{y}; \theta) = -(K-1)\log(\gamma \sum_{j \neq \bar{y}} p_\theta(j|x) + (1-\gamma)p_\theta(\hat{y}|x)), \ \hat{y} = \arg\max_{j \neq \bar{y}} p_c(j|x), \tag{10}$$

where $\gamma \in [0, 1]$ is a hyperparameter that controls the transition of learning. Recent studies found that adversarial training can degrade the generalization, and a trade-off between natural accuracy

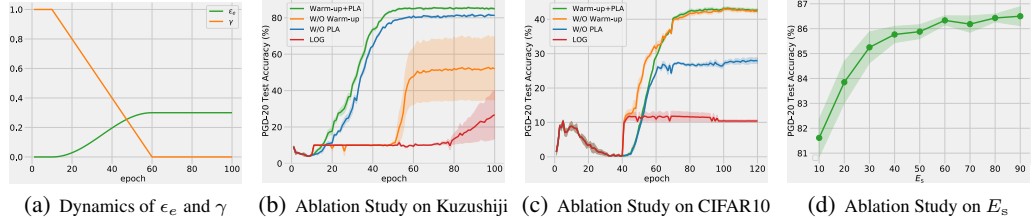

| (a) Dynamics of $\epsilon_e$ and $\gamma$ | (b) Ablation Study on Kuzushiji | (c) Ablation Study on CIFAR10 | (d) Ablation Study on $E_s$ |

Figure 3: (a) example of the dynamics of $\epsilon_e$ and $\gamma$ adopted on MNIST/Kuzushiji, where the warmup period is 10 epochs, and $E_s = 50$; (b) ablation study of the two components on Kuzushiji; (c) ablation study of the two components on CIFAR10; (d) PGD20 test accuracy on Kuzushiji w.r.t. the $E_s$.

and adversarial robustness exists [32]. To avoid the effect of such degradation and stabilize the optimization (e.g., a drastic shift of model predictions may occur due to the increasing adversarial budget), we use a simple exponential moving average to collect the dynamics of model predictions, and choose the label with the maximum probability as a predicted ordinary label. Specifically, given the model predictions $p_\theta(x)$ of the current epoch, the cashed model predictions $p_c(x)$ is updated as $p_c(x) = \beta p_c(x) + (1-\beta)p_\theta(x)$, where $p_c(x)$ is initialized with $\frac{(\mathbf{1} - e_{\bar{y}})^T}{K-1}$, $p_c(\bar{y}|x) = 0$, and $\beta = 0.9$.

## 5 Experiments

In this section, we empirically verify the effectiveness of our method on MNIST [22], Kuzushiji [8], CIFAR10 [20] and SVHN [28] datasets, and conduct ablation studies to understand the two attacks. More detailed information and the conducted experiments are presented in Appendix D.

**Setups.** 1) *General Setups*: for MNIST and Kuzushiji, we train a small convolutional neural network, as used in [41], for 100 epochs[3] with batch size of 256. For CIFAR-10 and SVHN, ResNet-18 is trained for 120 epochs with batch size of 128. The training sets of CIFAR-10 and SVHN are augmented with random cropping and horizontal flipping. 2) *Adversarial Training Setups*: we focus on the $L_\infty$ threat model. For MNIST and Kuzushiji, stochastic gradient descent (SGD) is used with learning rate of 0.01 and momentum of 0.9. PGD is used as the approximation method of solving inner maximization, with $\epsilon = 0.3$, $\alpha = 0.01$ and $k = 40$. For CIFAR10 and SVHN, SGD is used with weight decay of 5e-4 and momentum of 0.9. The initial learning rate is 0.01, and decayed by 10 at Epoch 30 and 60 of adversarial optimization. For the $L_\infty$ threat model, $\epsilon = 8/255$, $\alpha = 2/255$ and $k = 10$. 3) *Complementary Learning Setups*: following the settings in [18, 13], for MNIST and Kuzushiji, Adam optimizer is used with learning rate of 1e-3 and weight decay of 1e-4. For CIFAR10 and SVHN, SGD is used with weight decay of 5e-4 and momentum of 0.9. The learning rate is linearly increased to 0.01 within the first 5 epochs. More details are provided in Appendix D.1.

**Baseline.** 1). Two-stage method: it first trains a model for 50 epochs with the methods of complementary learning (i.e., LOG, which shows better performance and stability compared to others [13]). Then we re-label the complementary training set using the checkpoint with the best validation performance, and adversarially train another model using the re-labeled training set afterwards. 2). Conducting adversarial training directly with complementary losses, including FORWARD [39], FREE, NN [18], SCL_NL, SCL_EXP [7], EXP and LOG [13]. We evaluate the adversarial robustness using PGD20, CW30 and AutoAttack (AA) [9], and report the results of checkpoint with the best PGD20 test accuracy. All experiments are conducted with multiple random seeds (i.e., 1-3).

### 5.1 Performance Evaluation

The detailed results are reported in Table 1, where *Oracle* refers to the standard AT with ordinary labels, and it is just given for measuring the performance gap between AT with CLs and ordinary labels. For easy datasets (e.g., MNIST), the complementary losses are possible to obtain satisfying results, while most of them fail on complicated datasets. We would like to mention that the performance of

---

[3]Note that AT is nearly 10-40 times [24, 40] slower than the natural training due to the adversarial generation. Here we train 100 epochs which is kept the same as the most literature in AT [24, 41, 5, 29].

Table 1: Means (standard deviations) of natural and adversarial test accuracy.

| Dataset | Method | Natural | PGD | CW | AA |
|---------|--------|---------|-----|-----|-----|
| MNIST | Oracle | 99.46(±0.04) | 98.14(±0.04) | 97.45(±0.08) | 92.53(±0.23) |
| | Two-stage | 99.07(±0.02) | 97.44(±0.23) | 96.72(±0.21) | 92.06(±0.45) |
| | FORWARD [39] | 97.22(±1.13) | 93.76(±2.38) | 92.13(±2.87) | 85.41(±3.69) |
| | FREE [18] | 48.94(±26.04) | 38.02(±22.21) | 32.81(±19.63) | 22.68(±13.91) |
| | NN [18] | 68.48(±40.40) | 66.80(±39.22) | 66.25(±38.84) | 60.65(±38.04) |
| | SCL_NL [7] | 93.09(±7.35) | 87.07(±12.40) | 84.59(±14.51) | 75.98(±18.29) |
| | SCL_EXP [7] | 14.88(±4.99) | 14.34(±4.23) | 13.58(±3.16) | 10.47(±1.25) |
| | EXP [13] | 10.99(±0.50) | 10.99(±0.50) | 10.99(±0.50) | 10.99(±0.50) |
| | LOG [13] | 97.16(±0.64) | 93.38(±1.25) | 91.67(±1.39) | 84.88(±2.09) |
| | **Warm-up+PLA** | **99.22(±0.02)** | **97.73(±0.06)** | **97.11(±0.07)** | **92.37(±0.19)** |
| Kuzushiji | Oracle | 95.94(±0.15) | 90.01(±0.43) | 88.06(±0.96) | 70.63(±0.48) |
| | Two-stage | 89.75(±0.42) | 82.91(±1.01) | 80.21(±1.27) | 64.57(±1.79) |
| | FORWARD | 35.48(±27.96) | 29.84(±25.55) | 28.09(±24.37) | 22.01(±18.98) |
| | FREE | 16.17(±1.77) | 12.01(±0.55) | 9.33(±1.50) | 4.08(±1.60) |
| | NN | 10.00(±0.00) | 10.00(±0.00) | 10.00(±0.00) | 8.87(±1.60) |
| | SCL_NL | 40.83(±24.03) | 32.82(±22.88) | 29.93(±22.53) | 20.86(±19.74) |
| | SCL_EXP | 10.00(±0.00) | 10.00(±0.00) | 10.00(±0.00) | 8.21(±2.54) |
| | EXP | 10.00(±0.00) | 10.00(±0.00) | 10.00(±0.00) | 10.00(±0.00) |
| | LOG | 32.66(±25.50) | 26.87(±22.66) | 24.90(±21.20) | 18.78(±16.95) |
| | **Warm-up+PLA** | **91.60(±0.49)** | **85.88(±0.48)** | **83.74(±0.35)** | **68.75(±0.68)** |
| CIFAR10 | Oracle | 78.10(±0.36) | 47.35(±0.05) | 45.66(±0.22) | 43.47(±0.23) |
| | Two-stage | 64.98(±2.70) | 42.48(±0.92) | 39.90(±0.67) | 38.89(±0.46) |
| | FORWARD | 15.41(±0.85) | 13.58(±0.56) | 13.03(±0.34) | 12.94(±0.35) |
| | FREE | 11.60(±0.30) | 10.54(±0.23) | 10.45(±0.20) | 10.32(±0.26) |
| | NN | 11.79(±0.61) | 11.13(±0.56) | 11.14(±0.56) | 11.10(±0.53) |
| | SCL_NL | 14.45(±1.07) | 13.13(±1.03) | 12.84(±1.17) | 12.79(±1.18) |
| | SCL_EXP | 13.49(±1.76) | 12.55(±1.39) | 12.45(±1.35) | 12.38(±1.31) |
| | EXP | 10.97(±1.01) | 10.56(±0.60) | 10.38(±0.36) | 10.37(±0.34) |
| | LOG | 11.70(±1.06) | 11.04(±0.72) | 10.49(±0.59) | 10.44(±0.55) |
| | **Warm-up+PLA** | **65.88(±1.51)** | **43.29(±0.70)** | **41.30(±0.59)** | **40.28(±0.39)** |
| SVHN | Oracle | 91.95(±0.11) | 53.89(±0.11) | 50.59(±0.23) | 47.05(±0.22) |
| | Two-stage | **90.62(±0.27)** | 53.92(±0.33) | 50.86(±0.28) | 47.31(±0.11) |
| | FORWARD | 19.59(±0.00) | 19.59(±0.00) | 19.59(±0.00) | 19.68(±0.00) |
| | FREE | 19.59(±0.00) | 19.59(±0.00) | 19.59(±0.00) | 19.68(±0.00) |
| | NN | 19.59(±0.00) | 19.59(±0.00) | 19.59(±0.00) | 19.68(±0.00) |
| | SCL_NL | 19.59(±0.00) | 19.59(±0.00) | 19.58(±0.00) | 19.67(±0.00) |
| | SCL_EXP | 19.59(±0.00) | 19.59(±0.00) | 19.58(±0.00) | 19.68(±0.00) |
| | EXP | 19.60(±0.01) | 19.60(±0.02) | 19.56(±0.03) | 19.65(±0.04) |
| | LOG | 19.59(±0.00) | 19.60(±0.02) | 19.59(±0.00) | 19.68(±0.00) |
| | **Warm-up+PLA** | 90.50(±0.16) | **54.58(±0.10)** | **51.04(±0.04)** | **47.47(±0.13)** |

the two-stage and our methods slightly outperform that of the oracle on SVHN. We assume it may be attributed to the recent observation that noisy datasets (with a small portion of noisy labels) are not always detrimental to AT, and can even further boost the adversarial robustness [42].

## 5.2 Ablation Study

In this part, we conduct the ablation study to demonstrate the importance of the two components, i.e., Warm-up Attack and Pseudo-Label Attack in our method with the critical hyperparameter.

**Two Critical Components.** We investigate the roles of two components in our method by removing them separately. We report the results within three trials on Kuzushiji and CIFAR10 datasets as shown in the left-middle and right-middle panels of Figure 3. In summary, warm-up attack could stabilize

the optimization especially when the model capacity are not sufficient large, and pseudo-label attack is crucial for the achieving high adversarial robustness. Specifically, the warm-up attack achieves quite good performance on Kuzushiji, while limited improvements on CIFAR10. This is expected due to the difficulty of CIFAR10 without further informative attacks. Pseudo-label attack solves this problem by incorporating the informative model predictions, and hence achieves quite good performance on CIFAR10. However, it has an unstable result on Kusushiji, which is attributed to the limited model capacity. It is of great importance to achieve adversarial robustness given limited model capacity. Therefore, warm-up attack is still a crucial component for stabilizing the optimization.

**Hyperparameter.** We investigate the performance sensitivity to the key hyperparameter $E_{\mathrm{s}}$, which controls the growing speed of informative attacks. We report the results within three trials on Kuzushiji as shown in the right panels of Figure 3. The result shows a similar performance across a wide range of $E_{\mathrm{s}}$, which demonstrates that our method is not sensitive to the tuning of key hyperparameter.

## 6    Conclusion

To the best of our knowledge, this is the first work to study adversarial training (AT) with complementary labels (CLs), and to analyze its challenges from both theoretical and empirical perspectives. To solve them, we proposed a new learning strategy using gradually informative attacks, narrowing the performance gap between AT with CLs and ordinary labels. Our work shed light on the applications of AT to a more practical scenario (e.g., with imperfect supervision). In this paper, we focused on the standard AT algorithm, which formulates the problem as min-max optimization. We leave more advanced algorithms (e.g., TRADES) and other complementary settings (e.g., multiple CLs) to the future work. Moreover, CLs can be viewed as an extreme case of noisy labels with 100% noise rate, or partial labels where the labels other than the CL are the candidate set of labels. Hence, we hope our work could provide new insights for AT with various imperfect supervision in the ML community.

## Acknowledgments and Disclosure of Funding

JNZ and BH were supported by NSFC Young Scientists Fund No. 62006202, Guangdong Basic and Applied Basic Research Foundation No. 2022A1515011652, and RIKEN Collaborative Research Fund. BH was also supported by RGC Early Career Scheme No. 22200720, RGC Research Matching Grant Scheme No. RMGS2022_11_02 and No. RMGS2022_13_06, and HKBU CSD Departmental Incentive Grant. JFZ was supported by JSPS KAKENHI Grant Number 22K17955 and JST ACT-X Grant Number JPMJAX21AF, Japan. TLL was partially supported by Australian Research Council Projects DP180103424, DE-190101473, IC-190100031, DP-220102121, and FT-220100318. GN and MS were supported by JST AIP Acceleration Research Grant Number JPMJCR20U3, Japan. MS was also supported by the Institute for AI and Beyond, UTokyo. We would also like to thank the anonymous reviewers of NeurIPS 2022 for their constructive comments and recommendations.

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
