# Appendix

## A  Proofs

### A.1  Proof of Proposition 1

Given $\eta(x) = p(Y|X = x)$, $\bar{\eta}(x) = p(\bar{Y}|X = x)$, $\bar{\eta}(x) = Q^T\eta(x)$, $\ell(g(x)) = [\ell(1, g(x)), \ell(2, g(x)), \ldots, \ell(K, g(x))]$, and $e_i$ is the one-hot column vector with one on the $i_{\text{th}}$ entry, the unbiased risk estimator (URE) is derived as follows:

$$
\begin{aligned}
R(g; \ell) &= E_{(x,y)\sim\mathcal{D}}[\ell(y, g(x))] \\
&= E_{x\sim p(X)}E_{y\sim\eta(x)}[\ell(y, g(x))] \\
&= E_{x\sim p(X)}[\eta(x)^T\ell(g(x))] \\
&= E_{x\sim p(X)}[\bar{\eta}(x)^T(Q^{-1})\ell(g(x))] \\
&= E_{x\sim p(X)}E_{\bar{y}\sim\bar{\eta}(x)}[e_{\bar{y}}^T(Q^{-1})\ell(g(x))] \\
&= E_{(x,\bar{y})\sim\bar{\mathcal{D}}}[e_{\bar{y}}^T(Q^{-1})\ell(g(x))] \\
&= E_{(x,\bar{y})\sim\bar{\mathcal{D}}}[\bar{\ell}(\bar{y}, g(x))] \\
&= \bar{R}(g; \bar{\ell}).
\end{aligned}
\tag{11}
$$

We consider the single complementary label (SCL) setting with the uniform assumption, where complementary labels (CLs) are sampled uniformly from the candidate set $\mathcal{Y} \setminus \{y\}$, and Q is a transition matrix taking 0 on diagonals and $\frac{1}{K-1}$ on non-diagonals. The corrected loss $\bar{\ell}$ is then rewritten as

$$
\bar{\ell}(\bar{y}, g(x)) = e_{\bar{y}}^T(Q^{-1})\ell(g(x)) = -(K-1)\ell(\bar{y}, g(x)) + \sum_{j=1}^{K}\ell(j, g(x)).
\tag{12}
$$

### A.2  Proof of Proposition 2

For the general backward correction, based on Eq. 11, conducting adversarial training (AT) on the complementary risk is equivalent to that on the ordinary risk:

$$
\begin{aligned}
\min_{\theta} E_{x\sim p(X)} \max_{\tilde{x}\in\mathcal{B}_\epsilon[x]} E_{y\sim p(Y|X=x)}[\ell(y, g(\tilde{x}))] &= \min_{\theta} E_{x\sim p(X)} \max_{\tilde{x}\in\mathcal{B}_\epsilon[x]}[\eta(x)^T\ell(g(\tilde{x}))] \\
&= \min_{\theta} E_{x\sim p(X)} \max_{\tilde{x}\in\mathcal{B}_\epsilon[x]}[\bar{\eta}(x)^T(Q^{-1})\ell(g(\tilde{x}))] \\
&= \min_{\theta} E_{x\sim p(X)} \max_{\tilde{x}\in\mathcal{B}_\epsilon[x]} E_{\bar{y}\sim p(\bar{Y}|X=x)}[e_{\bar{y}}^T(Q^{-1})\ell(g(\tilde{x}))] \\
&= \min_{\theta} E_{x\sim p(X)} \max_{\tilde{x}\in\mathcal{B}_\epsilon[x]} E_{\bar{y}\sim p(\bar{Y}|X=x)}[\bar{\ell}(\bar{y}, g(\tilde{x}))].
\end{aligned}
\tag{13}
$$

In AT with ordinary labels, given $p(Y = y|X = x) \approx 1$, the empirical formulation is a proper estimator of the expected one:

$$
\begin{aligned}
\min_{\theta} E_{x\sim p(X)} \max_{\tilde{x}\in\mathcal{B}_\epsilon[x]} E_{y\sim p(Y|X=x)}[\ell(y, g(\tilde{x}))] &= \min_{\theta} E_{x\sim p(X)} \max_{\tilde{x}\in\mathcal{B}_\epsilon[x]} \sum_{j=1}^{K}[p(Y = j|X = x)\ell(j, g(\tilde{x}))] \\
&\approx \min_{\theta} E_{x\sim p(X)} \max_{\tilde{x}\in\mathcal{B}_\epsilon[x]}[\ell(y, g(\tilde{x}))] \\
&\approx \min_{\theta} \frac{1}{n}\sum_{i=1}^{n} \max_{\tilde{x}_i\in\mathcal{B}_\epsilon[x_i]}[\ell(y_i, g(\tilde{x}_i))].
\end{aligned}
\tag{14}
$$

While in AT with CLs, $p(\bar{Y} = j|X = x) = \sum_{k \neq j} p(\bar{Y} = j|Y = k)p(Y = k|X = x) \approx p(\bar{Y} = j|Y = y) \approx Q_{yj}$, which takes $\frac{1}{K-1}$ if $j \neq y$ and 0 otherwise. Therefore, the empirical formulation is not a proper estimator of the expected one on the SCL setting with the uniform assumption:

$$
\begin{aligned}
\min_\theta E_{x \sim p(X)} \max_{\tilde{x} \in \mathcal{B}_\epsilon[x]} E_{\bar{y} \sim p(\bar{Y}|X=x)}[\bar{\ell}(\bar{y}, g(\tilde{x}))] &= \min_\theta E_{x \sim p(X)} \max_{\tilde{x} \in \mathcal{B}_\epsilon[x]} \sum_{j=1}^{K}[p(\bar{Y} = j|X = x)\bar{\ell}(j, g(\tilde{x}))] \\
&\approx \min_\theta E_{x \sim p(X)} \max_{\tilde{x} \in \mathcal{B}_\epsilon[x]} \sum_{j \neq y}[Q_{yj}\bar{\ell}(j, g(\tilde{x}))] \\
&\neq \min_\theta E_{x \sim p(X)} \max_{\tilde{x} \in \mathcal{B}_\epsilon[x]}[\bar{\ell}(\bar{y}, g(\tilde{x}))] \\
&\neq \min_\theta \frac{1}{n} \sum_{i=1}^{n} \max_{\tilde{x}_i \in \mathcal{B}_\epsilon[x_i]}[\bar{\ell}(\bar{y}_i, g(\tilde{x}_i))].
\end{aligned}
\tag{15}
$$

Based on Eqs. 13, 14 and 15, the inequality holds between their empirical formulations:

$$
\min_\theta \frac{1}{n} \sum_{i=1}^{n} \max_{\tilde{x}_i \in \mathcal{B}_\epsilon[x_i]}[\ell(y_i, g(\tilde{x}_i))] \neq \min_\theta \frac{1}{n} \sum_{i=1}^{n} \max_{\tilde{x}_i \in \mathcal{B}_\epsilon[x_i]}[\bar{\ell}(\bar{y}_i, g(\tilde{x}_i))].
\tag{16}
$$

## B Loss Functions of Complementary Learning

Table 2 lists the popular loss functions in the literature of complementary learning, where $\bar{Y}_s$ is the set of multiple complementary labels (MCLs). FORWARD and FREE are the complementary loss functions derived by the forward correction [39] and the general backward correction [18] techniques, respectively. NN [18] is a non-negative correction method for fixing the overfitting issue of FREE in practice, by (lower) bounding the losses of all classes to 0. SCL_NL and SCL_EXP [7] are the methods for reducing empirical gradient variance by introducing a little bias. The above-mentioned are designed for the SCL setting, while EXP and LOG [13] are the bounded losses that could be used on both the SCL and MCLs settings.

Table 2: Loss Functions of Complementary Learning

| Method | Loss Function | SCL | MCLs |
|--------|---------------|-----|------|
| FORWARD | $-\log(\sum_{j \neq \bar{y}} T_{j\bar{y}} \cdot p_\theta(j|x))$ | ✓ | |
| FREE | $(K-1)\log p_\theta(\bar{y}|x) - \sum_{j=1}^{K} \log p_\theta(j|x)$ | ✓ | |
| NN | $\sum_{j=1}^{K} \max(0, \text{FREE}_j)$ | ✓ | |
| SCL_NL | $-\log(1 - p_\theta(\bar{y}|x))$ | ✓ | |
| SCL_EXP | $\exp(p_\theta(\bar{y}|x))$ | ✓ | |
| EXP | $\frac{K-1}{|\bar{Y}_s|} \exp(-\sum_{j \notin \bar{Y}_s} p_\theta(j|x))$ | ✓ | ✓ |
| LOG | $-\frac{K-1}{|\bar{Y}_s|} \log(\sum_{j \notin \bar{Y}_s} p_\theta(j|x))$ | ✓ | ✓ |

## C Empirical Analysis

We conduct experiments following the general setups (in Section 5), except that we set a fixed $\gamma = 0.5$ for simplicity of our method. We denote the AT with ordinary labels using cross-entropy as *oracle*.

### C.1 Gradient Norm

We adversarially train a model with several complementary losses separately on Kuzushiji. Figure 4 shows the (average and the corresponding single) $l_2$-Norm of stochastic gradients with respect to the model parameters on three random seeds. The results show the same observation (as in Section 4.2) that the gradient vanishing problem tends to occur at the early stage of adversarial optimization when using the complementary losses. Although FREE seems to have large gradient norm, it suffers from the huge empirical gradient variance problem [7] due to the fixed single complementary label given in practice, and hence inferior performance in both complementary learning and AT with CLs.

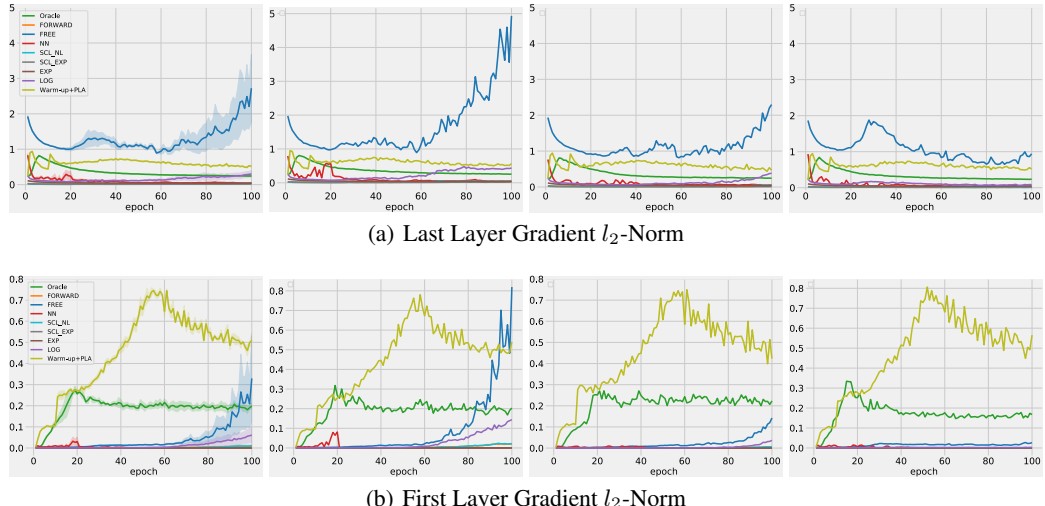

(a) Last Layer Gradient $l_2$-Norm

(b) First Layer Gradient $l_2$-Norm

Figure 4: (a) the (average and corresponding single) $l_2$-Norm of stochastic gradients of the last layer over three random trials; (b) the (average and corresponding single) $l_2$-Norm of stochastic gradients of the first layer over three random trials.

## C.2 Quality of Adversarial Examples

During adversarial optimization of *oracle* on MNIST and Kuzushiji, we generate adversarial examples[4] using various loss functions. We measure the quality of constructed adversarial examples through several metrics (e.g., cosine similarity, $l_1$-distance and model predictions), as shown in Figure 5. Moreover, we also show the results[5] on four randomly sampled instances from Kuzushiji in Figure 6, and the visualization of constructed adversarial examples in Figure 7. All results consistently show the same observations (as in Section 4.2) that existing complementary losses (as the objectives of AT with CLs) fails to generate high-quality adversarial examples.

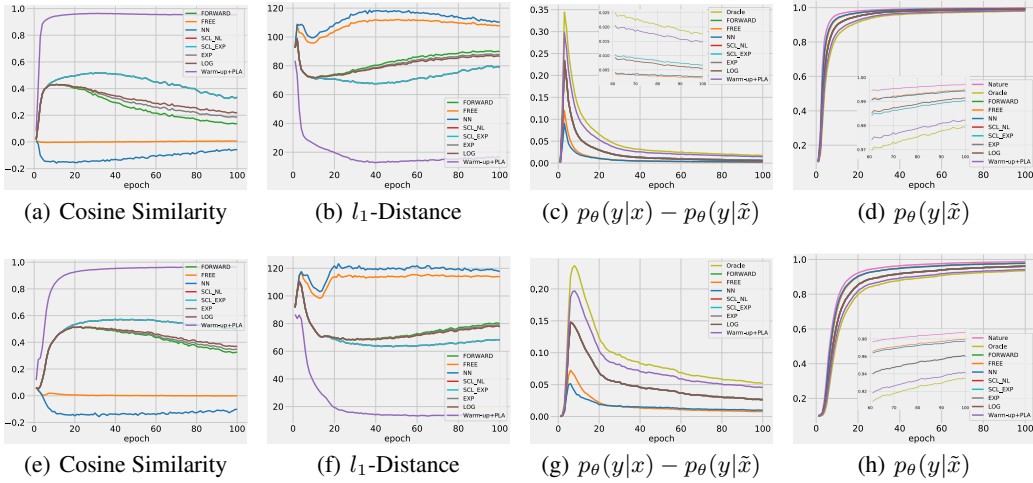

(a) Cosine Similarity  (b) $l_1$-Distance  (c) $p_\theta(y|x) - p_\theta(y|\tilde{x})$  (d) $p_\theta(y|\tilde{x})$

(e) Cosine Similarity  (f) $l_1$-Distance  (g) $p_\theta(y|x) - p_\theta(y|\tilde{x})$  (h) $p_\theta(y|\tilde{x})$

Figure 5: The *upper panels* shows the average results on MNIST, while the *lower panels* shows the average results on Kuzushiji, over all training samples. (a)/(e) the cosine similarity of gradient directions (i.e., $\text{sign}(\nabla_x \bar{\ell}(x, \bar{y}; \theta))$) with the oracle in the inner maximization; (b)/(f) the $l_1$-distance of the constructed adversarial examples with the oracle; (c)/(g) the difference of model predictions of natural data and adversarial data on the ordinary label; (d)/(h) the model predictions of natural data (i.e., Nature) and adversarial data on the ordinary label.

---

[4]Note that we only optimize the model using the ones generated by the oracle.

[5]We assume the extremely fluctuated curve of NN is attributed to its enforced non-negative loss correction of each class, which is the only difference between FREE and NN.

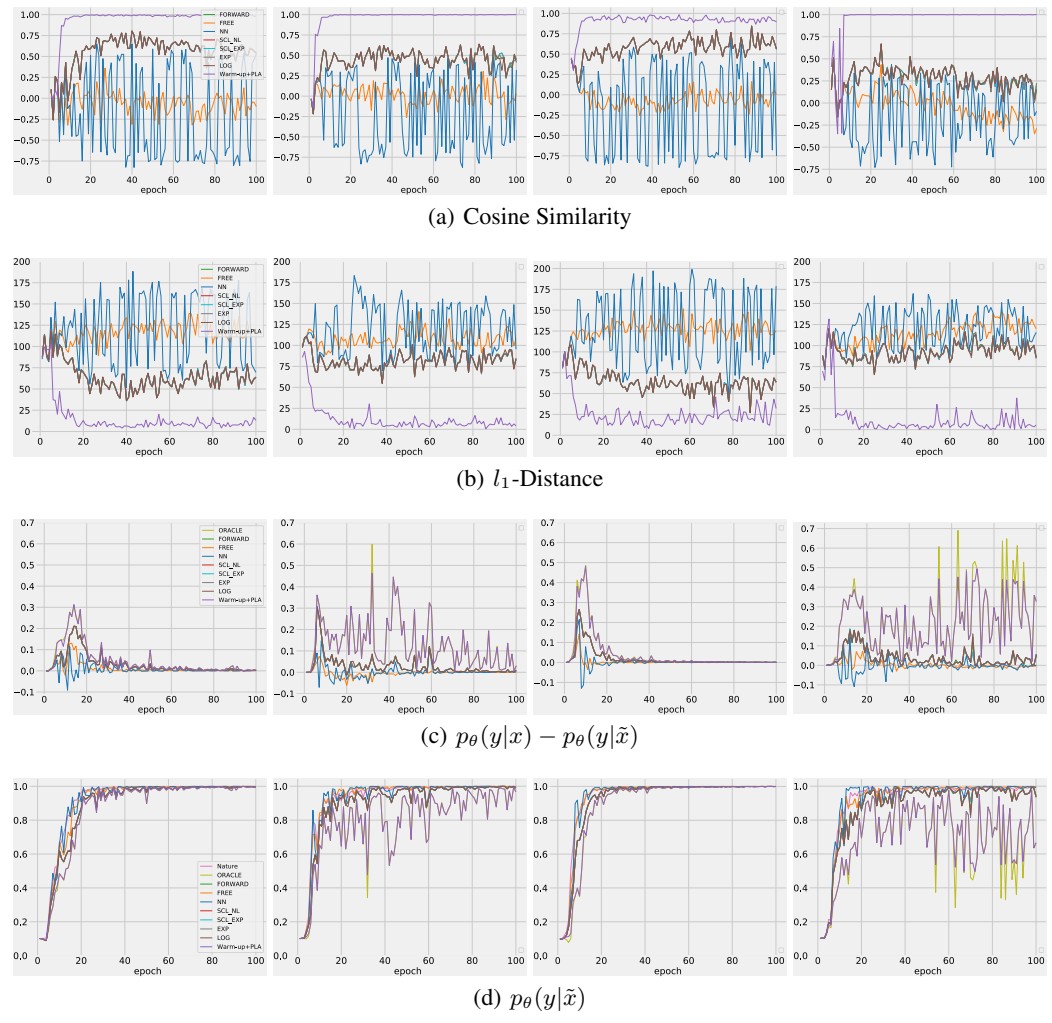

(a) Cosine Similarity

(b) $l_1$-Distance

(c) $p_\theta(y|x) - p_\theta(y|\tilde{x})$

(d) $p_\theta(y|\tilde{x})$

Figure 6: The results on four randomly sampled instances from Kuzushiji.

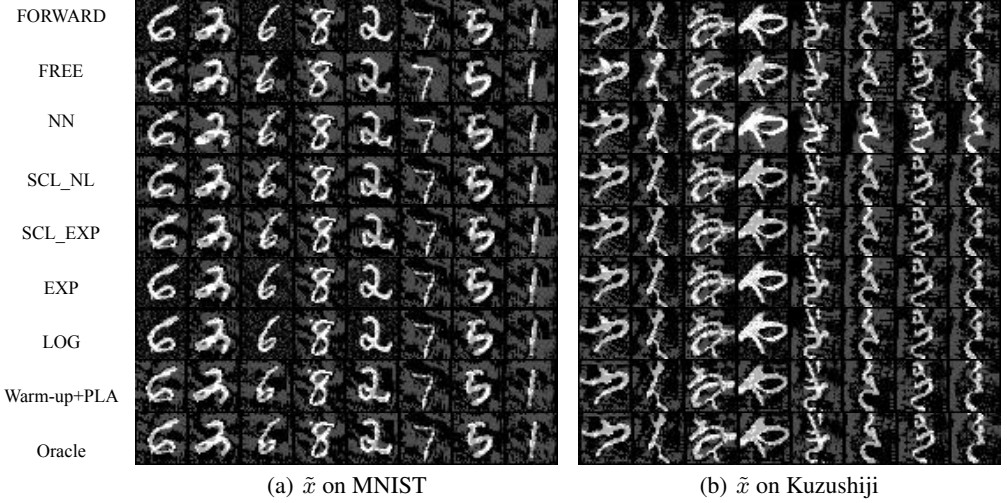

(a) $\tilde{x}$ on MNIST

(b) $\tilde{x}$ on Kuzushiji

Figure 7: The visualizations of adversarial examples constructed by various loss functions on MNIST (a) and Kuzushiji (b), given the same optimization model.

# D Experiments

## D.1 Extra Setups

**Setups for Performance Evaluation.** Following the adversarial training setups (in Section 5), for MNIST and Kuzushiji, our method starts with natural complementary learning for $E_i = 10$ epochs (i.e., an initial period to discard the less informative model predictions at the beginning of optimization). Then, the warm-up attack (Warm-up) and pseudo-label attack (PLA) are introduced with $E_s = 50$ (i.e., $\mathbb{T}(\epsilon_{\max} = 0.3, e - E_i, E_s = 50)$ and $\mathbb{T}(\gamma_{\max} = 1, e - E_i, E_s = 50)$, respectively). For CIFAR10 and SVHN, $E_i = 40$ and $E_s = 40$. An example of their dynamics adopted on MNIST/Kuzushiji is illustrated in the left panel of Figure 3. Moreover, a small adversarial budget could be viewed as an implicit way of data augmentation, while a large adversarial budget tends to sacrifice the generalization for adversarial robustness [32], hence we heuristically stop the update of cashed model predictions $p_c$ as long as the radius of epsilon ball exceeds $\epsilon/2$. For the baselines of AT with CLs, the two-stage method consists of a complementary learning phase and an AT phase, following the setups of complementary learning setups and AT setups (in Section 5), respectively. For the direct combinations of AT with CLs on MNIST/Kuzushiji, we set the learning rates of FORWARD [39], FREE, NN [18], SCL_NL, SCL_EXP [7], EXP and LOG [13] to 0.1, 0.001, 0.01, 0.1, 0.05, 0.01 and 0.01, respectively. For CIFAR10 and SVHN, their learning rates are set to 0.01. All experiments are conducted on NVIDIA GeForce RTX 3090.

**Setups for Ablation Study.** For Kuzushiji, all methods follow the previous settings (i.e., $E_i = 10$, $E_s = 50$, $\gamma_{\max} = 1$, $\epsilon_{\max} = 0.3$, $\alpha = 0.01$ and $K = 40$). For W/O Warm-up, $\gamma = \mathbb{T}(\gamma_{\max}, e - E_i, E_s)$, and $\epsilon_e$, $\alpha_e$, and $k_e$ are fixed to $\epsilon_{\max}$, $\alpha$ and $k$, respectively. Note that a suddenly shift of $\gamma$ from 1 to 0 would simply cause the failure of training. For W/O PLA, $\epsilon_e = \mathbb{T}(\epsilon_{\max}, e - E_i, E_s)$, $\alpha_e = \frac{\epsilon_e}{\epsilon_{\max}} \times \alpha$, $k_e = k$ and $\gamma = 1$. For CIFAR10, $E_i = 40$, $E_s = 40$, $\gamma_{\max} = 1$, $\epsilon_{\max} = 8/255$, $\alpha = 2/255$ and $k = 10$.

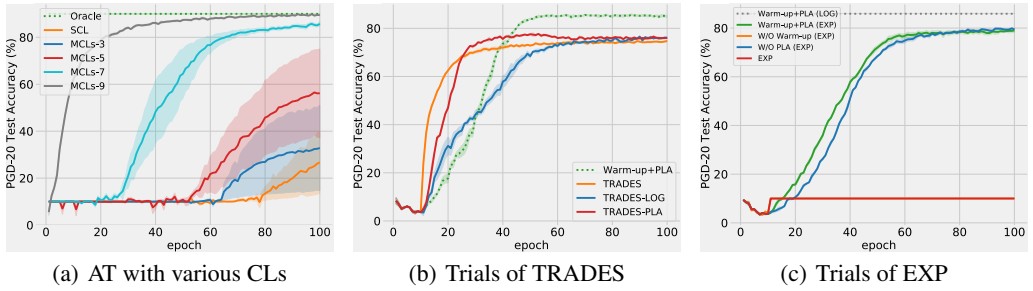

(a) AT with various CLs     (b) Trials of TRADES     (c) Trials of EXP

Figure 8: (a) a comparison of AT with different number of CLs; (b) a comparison with TRADES-based methods; (c) a comparison of our method combined with LOG and EXP.

## D.2 Empirical Justification for Proposition 2

Briefly, the proposition 2 (Eq. 8) and its proof (Appendix A.2) state that the intractable adversarial optimization is attributed to the limited CLs given in practice. Here, we conduct an experiment to justify it by giving MCLs (e.g., MCLs-3 represents that each data is assigned with three CLs) to each data. We conduct AT directly using the complementary loss on Kuzushiji following the previous setups. As the left panel of Figure 8 shown, the adversarial optimization tends to be stable and comparable to the oracle as the number of CLs increasing, which further justifies our proposition.

## D.3 Comparison with TRADES

TRADES [41] is an advanced AT algorithm without the needs of ordinary labels during the adversarial generation. Therefore, we adopt it to the AT with CLs, and conduct an experiment on Kuzushiji. Specifically, we generate adversarial examples (inner maximization) using their original implementation[6], while optimizing the training loss (outer minimization) using several variants: 1) *TRADES*: after

---

[6]https://github.com/yaodongyu/TRADES

Table 3: Means (standard deviations) of natural and adversarial test accuracy on Kuzushiji.

| Method | Natural | PGD | CW | AA |
|---|---|---|---|---|
| FORWARD [39] | 35.48($\pm$27.96) | 29.84($\pm$25.55) | 28.09($\pm$24.37) | 22.01($\pm$18.98) |
| +Warm-up | **91.39($\pm$0.60)** | **82.95($\pm$0.47)** | **79.69($\pm$0.52)** | **61.66($\pm$0.86)** |
| FREE [18] | 16.17($\pm$1.77) | 12.01($\pm$0.55) | 9.33($\pm$1.50) | 4.08($\pm$1.60) |
| +Warm-up | **83.19($\pm$1.39)** | **74.05($\pm$1.30)** | **70.48($\pm$1.17)** | **57.39($\pm$0.80)** |
| NN [18] | 10.00($\pm$0.00) | 10.00($\pm$0.00) | 10.00($\pm$0.00) | 8.87($\pm$1.60) |
| +Warm-up | **86.43($\pm$0.67)** | **77.85($\pm$0.98)** | **74.55($\pm$1.03)** | **59.84($\pm$0.73)** |
| SCL_NL [7] | 40.83($\pm$24.03) | 32.82($\pm$22.88) | 29.93($\pm$22.53) | 20.86($\pm$19.74) |
| +Warm-up | **91.92($\pm$0.29)** | 82.93($\pm$0.58) | 79.77($\pm$0.96) | 62.27($\pm$0.68) |
| +PLA | 39.64($\pm$32.18) | 32.18($\pm$30.82) | 28.37($\pm$30.97) | 21.58($\pm$23.01) |
| +Warm-up+PLA | 91.74($\pm$0.85) | **86.09($\pm$1.01)** | **83.77($\pm$1.03)** | **67.87($\pm$0.94)** |
| SCL_EXP [7] | 10.00($\pm$0.00) | 10.00($\pm$0.00) | 10.00($\pm$0.00) | 8.21($\pm$2.54) |
| +Warm-up | **89.09($\pm$0.58)** | 80.94($\pm$0.70) | 78.12($\pm$0.74) | 61.17($\pm$1.17) |
| +PLA | 10.00($\pm$0.00) | 10.00($\pm$0.00) | 10.00($\pm$0.00) | 10.00($\pm$0.00) |
| +Warm-up+PLA | 87.58($\pm$2.48) | **82.17($\pm$2.45)** | **80.25($\pm$2.41)** | **66.36($\pm$2.02)** |
| EXP [13] | 10.00($\pm$0.00) | 10.00($\pm$0.00) | 10.00($\pm$0.00) | 10.00($\pm$0.00) |
| +Warm-up | **89.18($\pm$0.21)** | **80.80($\pm$0.21)** | **77.30($\pm$0.35)** | 60.90($\pm$0.25) |
| +PLA | 10.00($\pm$0.00) | 10.00($\pm$0.00) | 10.00($\pm$0.00) | 10.00($\pm$0.00) |
| +Warm-up+PLA | 84.63($\pm$0.94) | 79.37($\pm$1.27) | 77.23($\pm$1.42) | **65.04($\pm$1.33)** |
| LOG [13] | 32.66($\pm$25.50) | 26.87($\pm$22.66) | 24.90($\pm$21.20) | 18.78($\pm$16.95) |
| +Warm-up | 91.31($\pm$0.53) | 82.77($\pm$0.33) | 80.31($\pm$0.25) | 62.23($\pm$0.42) |
| +PLA | 57.78($\pm$33.80) | 53.10($\pm$30.49) | 51.11($\pm$29.08) | 39.65($\pm$21.14) |
| +Warm-up+PLA | **91.60($\pm$0.49)** | **85.88($\pm$0.48)** | **83.74($\pm$0.35)** | **68.75($\pm$0.68)** |

the initial period $E_i = 10$, we optimize the model using the original TRADES loss with $1/\lambda = 1$, and regard predicted labels (i.e., $\arg\max_{j\neq\bar{y}} p_c(j|x)$) as ordinary labels; 2) *TRADES-LOG*: we directly use the complementary loss (i.e., LOG) to optimize the model; 3) *TRADES-PLA*: we still regard predicted labels as ordinary labels, but using Eq. 10 as the objective of outer minimization. As the middle panel of Figure 8 shown, the results are not comparable to ours. A specifically designed loss function may be needed, and we would leave it to the future work.

### D.4 Gradually Informative Attacks with Other Complementary Losses

We also try to incorporate our proposed method into other complementary losses (i.e., EXP [13], SCL_NL and SCL_EXP [7]), which could be rewritten as

$$\bar{\ell}_{\text{EXP}}(x,\bar{y};\theta) = (K-1)\exp(-\gamma \sum_{j\neq\bar{y}} p_\theta(j|x) - (1-\gamma)p_\theta(\hat{y}|x)), \ \hat{y} = \arg\max_{j\neq\bar{y}} p_c(j|x). \quad (17)$$

$$\bar{\ell}_{\text{SCL\_NL}}(x,\bar{y};\theta) = -\log(\gamma(1-p_\theta(\bar{y}|x)) + (1-\gamma)p_\theta(\hat{y}|x)), \ \hat{y} = \arg\max_{j\neq\bar{y}} p_c(j|x). \quad (18)$$

$$\bar{\ell}_{\text{SCL\_EXP}}(x,\bar{y};\theta) = \exp(\gamma(p_\theta(\bar{y}|x)) - (1-\gamma)p_\theta(\hat{y}|x)), \ \hat{y} = \arg\max_{j\neq\bar{y}} p_c(j|x). \quad (19)$$

Following the previous setups, we conduct experiments on Kuzushiji. Taking EXP as an example, as shown in the right panel of Figure 8, our method (i.e., Warm-up+PLA (EXP)) could still greatly ease the adversarial optimization and improve the adversarial robustness, though it is not comparable with the one with LOG. The performance gap between LOG and EXP may be attributed to the curvatures and optimizers. We summarize the comprehensive ablation study results in Table 3[7]

---

[7]The Warm-up is readily to combine with any complementary losses, while the PLA needs further design to combine with them (e.g., FORWARD, FREE and NN).

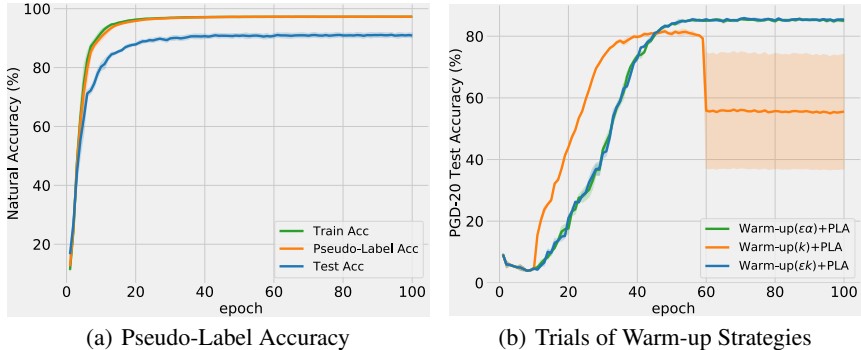

| (a) Pseudo-Label Accuracy | (b) Trials of Warm-up Strategies |

Figure 9: (a) accuracy dynamics of pseudo-labels (obtained by exponential moving average) during adversarial optimization with CLs; (b) a comparison with various strategies of Warm-up attack.

The results (e.g., Figure 3 (b-c) and Table 3) demonstrate the benefits of our proposed unified framework, which naturally combines and adaptively controls the two indispensable components throughout the adversarial optimization with CLs. The two indispensable components have different motivations and underlying principles with other related literature [16, 23], and are closely related to the unique challenges identified in AT with CLs. Without either of them and a reasonable scheduler, the adversarial optimization with CLs will be extremely unstable and result in a model with poor robustness, or even failure.

## D.5 Multiple Complementary Labels

Our proposed method can be easily adopted to the MCLs setting with the loss form of

$$\bar{\ell}_{\text{MCLs}}(x, \bar{Y}_s; \theta) = -\frac{(K-1)}{|\bar{Y}_s|} \log(\gamma \sum_{j \notin \bar{Y}_s} p_\theta(j|x) + (1-\gamma)p_\theta(\hat{y}|x)), \ \hat{y} = \arg\max_{j \notin \bar{Y}_s} p_c(j|x), \quad (20)$$

where $\bar{Y}_s$ is the set of MCLs. In Table 4, we conduct experiments on Kuzushiji and CIFAR10, following the data distribution described in [13] and the same setups described in Section 5. As the results shown, in contrast to the SCL setting, the label information is naturally enhanced by MCLs, which leads to the better robustness. Incorporating novel insights of partial labels or noisy labels may further boost the robustness. Due to the above difference, we would leave it to the future work.

Table 4: Means (standard deviations) of natural and adversarial test accuracy.

| Dataset | Method | Natural | PGD | CW | AA |
|---|---|---|---|---|---|
| | Oracle | 95.94($\pm$0.15) | 90.01($\pm$0.43) | 88.06($\pm$0.96) | 70.63($\pm$0.48) |
| Kuzushiji-MCLs | Two-stage | 93.53($\pm$0.84) | 86.52($\pm$1.68) | 84.19($\pm$2.21) | 65.91($\pm$2.69) |
| | EXP | 10.00($\pm$0.00) | 10.00($\pm$0.00) | 9.99($\pm$0.01) | 7.25($\pm$3.89) |
| | LOG | 77.83($\pm$17.72) | 68.87($\pm$19.21) | 64.69($\pm$20.63) | 50.04($\pm$18.14) |
| | Warm-up+PLA | **95.61($\pm$0.02)** | **89.89($\pm$0.17)** | **88.02($\pm$0.25)** | **71.48($\pm$0.12)** |
| | Oracle | 78.10($\pm$0.36) | 47.35($\pm$0.05) | 45.66($\pm$0.22) | 43.47($\pm$0.23) |
| CIFAR10-MCLs | Two-stage | 77.11($\pm$0.10) | **47.58($\pm$0.13)** | **45.60($\pm$0.09)** | **43.63($\pm$0.08)** |
| | EXP | 37.01($\pm$1.65) | 29.31($\pm$0.93) | 28.05($\pm$0.77) | 27.42($\pm$0.70) |
| | LOG | 52.02($\pm$0.50) | 36.72($\pm$0.04) | 33.79($\pm$0.20) | 32.53($\pm$0.19) |
| | Warm-up+PLA | **82.33($\pm$0.18)** | 46.44($\pm$0.03) | 45.00($\pm$0.27) | 42.56($\pm$0.32) |

## D.6 Empirical Evaluation of Pseudo-Label Accuracy

To avoid the effect of (natural complementary learning) warmup period on the analysis of accuracy dynamics of pseudo-labels, we set $E_{\text{i}} = 0$ while keeping other setups fixed (e.g., the $\epsilon$ is increased

Table 5: Accuracy of pseudo-labels (%) w.r.t. the slowly increased $\epsilon$ ball in each epoch.

| Epoch | 1 | 2 | 3 | 4 | 5 |
|---|---|---|---|---|---|
| Acc. | 12.52($\pm$0.42) | 25.44($\pm$2.26) | 46.78($\pm$4.27) | 61.57($\pm$4.53) | 71.81($\pm$6.13) |
| $\epsilon_e$ | 0.0003 | 0.0012 | 0.0027 | 0.0047 | 0.0073 |
| $\epsilon_e/\epsilon_{\max}$ | 0.10% | 0.39% | 0.89% | 1.57% | 2.45% |
| Epoch | 6 | 7 | 8 | 9 | 10 |
| Acc. | 79.70($\pm$4.20) | 85.27($\pm$1.63) | 87.55($\pm$1.05) | 89.00($\pm$1.05) | 90.39($\pm$1.15) |
| $\epsilon_e$ | 0.0105 | 0.0143 | 0.0186 | 0.0234 | 0.0286 |
| $\epsilon_e/\epsilon_{\max}$ | 3.51% | 4.76% | 6.18% | 7.78% | 9.55% |

Table 6: Accuracy of pseudo-labels (%) w.r.t. the rapidly increased $\epsilon$ ball in each epoch.

| Epoch | 1 | 2 | 3 | 4 | 5 |
|---|---|---|---|---|---|
| Acc. | 11.08($\pm$0.04) | 12.66($\pm$1.21) | 12.93($\pm$1.48) | 12.98($\pm$1.52) | 13.0($\pm$1.55) |
| $\epsilon_e$ | 0.0073 | 0.0286 | 0.0618 | 0.1036 | 0.1500 |
| $\epsilon_e/\epsilon_{\max}$ | 2.45% | 9.55% | 20.61% | 34.55% | 50.00% |
| Epoch | 6 | 7 | 8 | 9 | 10 |
| Acc. | 13.02($\pm$1.56) | 13.02($\pm$1.56) | 13.02($\pm$1.56) | 13.02($\pm$1.56) | 13.01($\pm$1.55) |
| $\epsilon_e$ | 0.1964 | 0.2382 | 0.2714 | 0.2927 | 0.3000 |
| $\epsilon_e/\epsilon_{\max}$ | 65.45% | 79.39% | 90.45% | 97.55% | 100.00% |

from 0 to 0.3 within the first $E_s = 50$ epochs following the scheduler described in Section 4.3). We conduct experiments on Kuzushiji, and show the result in the left panel of Figure 9 and Table 5. We observe the accuracy rises up rapidly at the early stage of adversarial optimization with CLs, which demonstrates that a small epsilon ball (e.g., $\epsilon$ is increased from 0 to 0.029 within the first 10 epochs) is helpful for the formation of a discriminative model that tends to assign high confidence to the ordinary labels. We also try a relatively large epsilon ball at the early stage of adversarial optimization (i.e., by only modifying $E_s = 10$). As shown in Table 6, the model fails to assign high confidence to the ordinary label in such a case, and even may fail the adversarial optimization.

### D.7 Ablation Study on Strategies of Warm-up Attack

In the main paper, we implement the Warm-up attack (i.e., Warm-up($\epsilon\alpha$)+PLA) by controlling the radius of the epsilon ball $\epsilon$ and step size $\alpha$, while fixing the number of attack steps $k$. Here, we conduct experiments on Kuzushiji, with two more strategies of Warm-up attack: 1) Warm-up($k$)+PLA: we only change the number of attack steps based on the same scheduler described in Section 4.3; 2) Warm-up($\epsilon k$)+PLA: similar to the original implementation, we proportionally increase $k$ ($k_e = \lceil \frac{\epsilon_e}{\epsilon} k \rceil$) instead of the step size based on the dynamics of epsilon ball. As the middle panel of Figure 9 shown, the performace of Warm-up($\epsilon k$)+PLA is comparable to that of Warm-up($\epsilon\alpha$)+PLA. However, Warm-up($k$)+PLA tends to unstabilize the adversarial optimization in some runs, which may be caused by the huge gradient variance in the big epsilon ball as our empirical analysis in Section 4.2. In practice, Warm-up($\epsilon k$)+PLA may be a good choice considering both the performance and computational efficiency.

# E  Broader Impact

Adversarial training (AT) is one of the most effective defensive methods against human-imperceptible perturbations. Although AT (with perfect supervision) has been thoroughly studied recently, the study of AT with imperfect supervision is an unexplored yet significant direction. In this paper, we study AT with complementary labels (CLs). From both theoretical and empirical perspectives, we identity the challenges of AT with CLs. To solve the problems, we propose a new learning strategy using gradually informative attacks, and reduce the performance gap of AT with ordinary labels and CLs. Since we introduce extra operations on the basis of AT with CLs for addressing the algorithmic issues, the computation cost may not be friendly to our environment. In this paper, we mainly focus on the SCL with the uniform assumption. Other complementary settings (e.g., biased CLs or MCLs) are worthy of attention as well. Although we take a step forward in AT with imperfect supervision, there are many other practical tasks needed to be explored further (e.g., noisy labels or partial labels).