# OpenReview forum: "Adversarial Training with Complementary Labels: On the Benefit of Gradually Informative Attacks"
_NeurIPS.cc/2022/Conference — NeurIPS 2022 Accept_

### Official Review · Reviewer_Jbnu · 2022-07-10

**Rating:** 6
**Confidence:** 4
**Soundness:** 3 good
**Presentation:** 3 good
**Contribution:** 3 good

**Summary:**

This paper aims to propose an effective adversarial training (AT) method for the scenario where the imperfect supervision is available. More specifically, the paper shows that when the complementary label (CL) (i.e. label for non-groundtruth class) are available, a direct combination of AT and CL will fail. To address this limitation, two attack approaches, warm-up attack and pseudo-label attack, are proposed. The former gradually increases the attack budget over the training epoch and the latter uses the pseudo-label of the model prediction to generate the adversarial example. Experiments demonstrate the effectiveness of the proposed attack methods under the imperfect supervision scenario.

**Questions:**

The author is suggested to address the concern in the weakness section, especially the novelty of this work (e.g. the proposed 2 attacks).

**Limitations:**

Yes, the author has provided the checklist and the discuss the broader impact in the supplemental material.

**Strengths And Weaknesses:**

Strength

1.	This work first proposed adversarial training together with complementary labels. From my best understanding, this is the first work to study this problem.

2.	The motivation of studying adversarial training in imperfect data scenario is strong and practical.

3.	Algorithm 1 is clear and simple to follow.

Weakness

1.	In L24, the author mentioned that the adversarial training (AT) with imperfect supervision has received less attention. While this work mainly studies the use of complementary labels, there are papers like [a], which studies AT with noisy label. The author is suggested to compare the proposed method with [a], which both study the AT in imperfect data regime.

2.	The novelty of the proposed method is a concern of this work. The warm-up technique has been widely used in deep learning. For example, [b] studies the warm-up technique used to adjust the learning rate.  [c] is somewhat similar to the proposed Warm-up Attack, where the attack budget gradually increases (See section 5 of [c]). On the other hand, the pseudo-label attack mainly stabilizes the adversarial example generation by using the pseudo-label from model prediction. This has been widely used, for example, when applying adversarial training under semi-supervised learning [d] (See Meta-Algorithm 1 in section 4 of [d]). Since both proposed attack methods appear in the literature, the author is suggested to highlight the novelty of the proposed methods.

3.	Figure 3 only conducts ablation on the LOG method, does the improvement of adding Warm-up Attack and Pseudo-label Attack applied to other complementary learning baselines?



[a] Understanding the Interaction of Adversarial Training with Noisy Labels
[b] A closer look at deep learning heuristics: Learning rate restarts, warmup and distillation
[c] On the Loss Landscape of Adversarial Training: Identifying Challenges and How to Overcome Them
[d] Unlabeled Data Improves Adversarial Robustness

---

> ### Author Response · Authors · 2022-08-01
> **Response to Reviewer Jbnu: Part 1**
>
> Thanks for reviewing our paper, with valuable comments and suggestions. Here are our detailed responses to your suggestions/questions.
>
> **Q1: Compare with the literature considering the interaction of AT with Noisy Labels [a].**
>
> **A1:** Thanks for your suggestion. However, we would like to mention that paper [a] does not target for adversarial training under imperfect supervision. It studied and explored the interaction of AT with Noisy Labels (NLs). That paper found AT itself is more robust than natural training and the number of PGD steps needed to attack a point can be used together with the small-loss trick (i.e., **they use AT to help label noise**). Except for the main difference, combining AT with robust learning methods for noisy labels is also not compatible with our problem setting. Below, we briefly design a new algorithm in a three-stage way by leveraging the insights in [a], and conduct experiments to evaluate its performance on AT with CLs:
>
> - First of all, [a] did not focus on solving the adversarial training on noisy datasets. It explored the interactions of adversarial training and noisy labels (e.g., the smoothing effect of adversarial training). Note that [a] did not propose a specific method targeted for conducting AT under imperfect supervision. Here, we leverage the specific empirical insights from noisy labels [1], that is, clean samples tend to have small losses and small pgd steps proposed in [a]. However, it is not applicable to complementary learning (CL is a special case of NL with a 100% noise rate). Therefore, it is hard to design a one-stage method leveraging their empirical insight from NLs, for AT with CLs.
> - Moreover, if we consider the two-stage baseline, it is possible to add a new stage that conducts sample selection based on the empirical insights from [a], after the first (complementary learning) stage. In this way, the newly designed algorithm consists of three stages (i.e., complementary learning -> sample selection -> vanilla AT). Specifically, it first converts the given complementary dataset $(x, \bar{y})$ into a noisy dataset $(x, \tilde{y})$ through the technique of complementary learning, then leverages the insights of [a] to select potential clean samples on the generated noisy dataset, and finally conducts vanilla AT with the selected samples. However, the sample selection phase usually requires knowing the noise rate in advance, which is impractical/impossible to accurately estimate in our settings (since we do not have any information about the ordinary label).
> - However, we still design the algorithm in a three-stage way and conduct an experiment on Kuzushiji following the setting of the two-stage baseline (as described in section 5 and Appendix D.1), by assuming the ordinary label of each data $(x, y, \bar{y})$ is known in advance. During the sample selection phase, we set the estimated noise rate as 1-$ACC_{Tr}$, where $ACC_{Tr}$ is the natural training accuracy of the model in the last epoch. But note that this is an unfair setting since only an *oracle* is able to know that. As Table 1 shown, we do not observe superior performance in the experiment.
> - Actually, it is expected that the performance of the designed three-stage method is similar to that of the two-stage baseline, since there are no essential differences between the label correction (e.g., the first stage) and the following sample selection (i.e., the second stage). The generated noisy dataset $(x, \tilde{y})$ (after the first stage) is instance-dependent noise (IDN) rather than class-conditional noise (CCN) on which most of the methods in the literature of CLs / NLs are focusing.
>
> Overall, we summarize the comparison results in Table 1.
>
> **Table 1.** Performance Comparison.
>
> |            Method             |       Natural        |         PGD          |          CW          |          AA          |
> | :---------------------------: | :------------------: | :------------------: | :------------------: | :------------------: |
> |      Two-stage baseline       |   89.75($\pm$0.42)   |   82.91($\pm$1.01)   |   80.21($\pm$1.27)   |   64.57($\pm$1.79)   |
> | Three-stage with [a] insights |   89.00($\pm$0.67)   |   82.37($\pm$1.25)   |   80.29($\pm$2.00)   |   64.09($\pm$1.34)   |
> |          Warm-up+PLA          | **91.60($\pm$0.49)** | **85.88($\pm$0.48)** | **83.74($\pm$0.35)** | **68.75($\pm$0.68)** |
>
> [a] Understanding the Interaction of Adversarial Training with Noisy Labels.
>
> [1] A closer look at memorization in deep networks.

---

> ### Author Response · Authors · 2022-08-01
> **Response to Reviewer Jbnu: Part 2**
>
> **Q2: About the novelty of this work. Both proposed attack methods appear in the literature.**
>
> **A2:** There are many facets to evaluate the novelty of the work. In our opinion, the main novelty of our work is the proposed new setting (AT with CLs), the proposed unified framework based on our theoretical and empirical analysis, and the potential insights for future research work on AT with various imperfect supervision. Complementary learning illustrates the possibility of training ordinary classifiers even if all the labels given for training are wrong. We take one step further by taking robustness benefits from adversarial training with ordinary labels, and tackling the unique challenges of AT with CLs. The studied new problem setting itself is of scientific interests to both research areas of weakly supervised learning and adversarial learning.
>
> As for the technical novelty, first, the proposed method is conceptually novel since it is naturally based on the unique challenges identified in AT with CLs, which have both theoretical and empirical insights for the new research problem. It has a different motivation and underlying principle from other literature. Different from simply labeling the unlabeled data [d] in a one-shot manner (**the same as the two-stage baseline**), our Pseudo-label Attack (PLA) further utilizes the dynamics of adversarial optimization with CLs (e.g., weight between CL and PL according to the learning status). Different from improving training stability in the large batch training in [b] or escaping from the suboptimal random initialization in [c], our Warm-up Attack (Warm-up) is for easing the adversarial optimization with CLs considering the CLs information in different epsilon balls.
>
> Second, except for the high-level differences, our proposed method has technical differences from the [b-d]. As for PLA, [d] only considers the pseudo-label (PL) generated by the fixed pre-trained model as below, which is not optimized further.
>
> > ```Meta-Algorithm 1 in section 4 of [d]:```
> >
> > 1. Learn a model using standard training;
> > 2. Generate pseudo-label for unlabeled data using the fixed standard model;
>
> While our method generates PL based on the cashed probability to improve optimization stability with CLs, and adaptively considers both PLs and CLs during the optimization process (refer to Eq.(10)). As for Warm-up, we also consider different strategies (refer to Section 4.3 and Appendix D.7) that can achieve our unique objective instead of only controlling the radius of the epsilon ball or learning rate that is closely related to their corresponding research problem in [b,c].
>
> More importantly, our proposed method is a *unified framework* that integrates the two critical components together in a quite natural way and controls them adaptively throughout the adversarial optimization with CLs. Both two components are indispensable, and are proposed to solve the corresponding challenges of AT with CLs, based on our theoretical and empirical analysis. We kindly refer to our response to Q3 for the ablation study of each component combined with several complementary learning baselines, without either of the two components and reasonable scheduling of them, the adversarial optimization with CLs will be extremely unstable and result in a model with poor robustness, or even failure (see also in Section 5.2 and Appendix D.4).
>
> [b] A closer look at deep learning heuristics: Learning rate restarts, warmup and distillation.
>
> [c] On the Loss Landscape of Adversarial Training: Identifying Challenges and How to Overcome Them.
>
> [d] Unlabeled Data Improves Adversarial Robustness.

---

> ### Author Response · Authors · 2022-08-01
> **Response to Reviewer Jbnu: Part 3**
>
> **Q3: Demonstrate the improvements of adding Warm-up Attack and Pseudo-label Attack with other complementary learning baselines**
>
> **A3:** To demonstrate the effectiveness of adding Warm-up (Warm-up) Attack and Pseudo-labels Attack (PLA), we conduct the experiments with other complementary learning baselines. To be specific, we run the experiments on Kuzushiji dataset following the settings described in the main paper. We summarize the results (mean with standard deviations) within 3 runs in Table 2.
>
> **Table 2.** Performance of adding Warm-up and PLA with other complementary learning baselines.
>
> |                       |       Natural        |         PGD          |          CW          |          AA          |
> | :-------------------- | :------------------: | :------------------: | :------------------: | :------------------: |
> | SCL_NL                |  40.83($\pm$24.03)   |  32.82($\pm$22.88)   |  29.93($\pm$22.53)   |  20.86($\pm$19.74)   |
> | SCL_NL+Warm-up        | **91.92($\pm$0.29)** |   82.93($\pm$0.58)   |   79.77($\pm$0.96)   |   62.27($\pm$0.68)   |
> | SCL_NL+PLA            |  39.64($\pm$32.18)   |  32.18($\pm$30.82)   |  28.37($\pm$30.97)   |  21.58($\pm$23.01)   |
> | Warm-up+PLA (SCL_NL)  |   91.74($\pm$0.85)   | **86.09($\pm$1.01)** | **83.77($\pm$1.03)** | **67.87($\pm$0.94)** |
> |                       |                      |                      |                      |                      |
> | SCL_EXP               |   10.00($\pm$0.00)   |   10.00($\pm$0.00)   |   10.00($\pm$0.00)   |   8.21($\pm$2.54)    |
> | SCL_EXP+Warm-up       | **89.09($\pm$0.58)** |   80.94($\pm$0.70)   |   78.12($\pm$0.74)   |   61.17($\pm$1.17)   |
> | SCL_EXP+PLA           |   10.00($\pm$0.00)   |   10.00($\pm$0.00)   |   10.00($\pm$0.00)   |   10.00($\pm$0.00)   |
> | Warm-up+PLA (SCL_EXP) |   87.58($\pm$2.48)   | **82.17($\pm$2.45)** | **80.25($\pm$2.41)** | **66.36($\pm$2.02)** |
> |                       |                      |                      |                      |                      |
> | EXP                   |   10.00($\pm$0.00)   |   10.00($\pm$0.00)   |   10.00($\pm$0.00)   |   10.00($\pm$0.00)   |
> | EXP+Warm-up           | **89.18($\pm$0.21)** | **80.80($\pm$0.21)** | **77.30($\pm$0.35)** |   60.90($\pm$0.25)   |
> | EXP+PLA               |   10.00($\pm$0.00)   |   10.00($\pm$0.00)   |   10.00($\pm$0.00)   |   10.00($\pm$0.00)   |
> | Warm-up+PLA (EXP)     |   84.63($\pm$0.94)   |   79.37($\pm$1.27)   |   77.23($\pm$1.42)   | **65.04($\pm$1.33)** |
> |                       |                      |                      |                      |                      |
> | LOG                   |  32.66($\pm$25.50)   |  26.87($\pm$22.66)   |  24.90($\pm$21.20)   |  18.78($\pm$16.95)   |
> | LOG+Warm-up           |   91.31($\pm$0.53)   |   82.77($\pm$0.33)   |   80.31($\pm$0.25)   |   62.23($\pm$0.42)   |
> | LOG+PLA               |  57.78($\pm$33.80)   |  53.10($\pm$30.49)   |  51.11($\pm$29.08)   |  39.65($\pm$21.14)   |
> | Warm-up+PLA (LOG)     | **91.60($\pm$0.49)** | **85.88($\pm$0.48)** | **83.74($\pm$0.35)** | **68.75($\pm$0.68)** |
>
> Similar to the results shown in Figure 8(c) of Appendix D.4, the improvement of adding Warm-up Attack and Pseudo-label Attack can be also found in other complementary learning baselines. We would also like to mention that *only with Warm-up* seems to be empirically effective for datasets like MNIST and Kuzushiji. However, based on our previous experiments, all complementary learning baselines equip only with Warm-up result in unsatisfactory robustness on the CIFAR-10 datasets (e.g., LOG+Warm-up can only achieve 28.25% of PGD20 test accuracy on CIFAR10 as shown in Figure 3(c)). Also, if *only with PLA*, the adversarial optimization tends to be unstable or even fail (e.g., the results for SCL+EXP+PLA and EXP+PLA in Table 2). The above two facts further verify the benefits of the proposed *unified framework* (Warm-up+PLA), which can achieve consistently better performance as demonstrated in our experiments.
>
> We will update the comprehensive results and corresponding analysis in Appendix D.4.

---

> ### Author Response · Authors · 2022-08-06
> **Would you mind checking our response? Thanks!**
>
> Dear Reviewer Jbnu,
>
> We appreciate your efforts in reviewing our paper. Would you mind checking our response, and is there any unclear point so that we could further clarify?
>
> Best regards,
> Authors

---

> ### Author Response · Authors · 2022-08-08
> **[Last Two days Reminder] Would you mind confirming if you have further questions? Thanks!**
>
> Dear Reviewer Jbnu,
>
> We appreciate your efforts in reviewing our paper. We have addressed all your questions in detail. Would you mind checking our response, and confirming if you have further questions?
>
> Best regards,
> Authors

---

> ### Author Response · Authors · 2022-08-08
> **Author Rebuttal Acknowledgement**
>
> Dear Reviewer Jbnu,
>
> Would you mind acknowledging our rebuttal? As the discussion due is approaching, if you still have some questions, let us discuss in the openreview system.
>
> Best regards,
>
> Authors

---

> ### Comment · Reviewer_Jbnu · 2022-08-08
> **Response to the rebuttal**
>
> Thanks the author for the through response and sorry for the late reply. The author has addressed most of my concerns, so I would raise my initial score and recommend this work.

---

### Official Review · Reviewer_u259 · 2022-07-10

**Rating:** 7
**Confidence:** 4
**Soundness:** 4 excellent
**Presentation:** 4 excellent
**Contribution:** 2 fair

**Summary:**

This paper proposes to address a new problem: adversarial training with Complementary Labels (CLs). A naive combination of Adversarial training and CLs fails to yield good performance. The authors identified the problem of this naive combination and propose to use Warm-up Attack and Pseudo Label Attack to address these problems. The proposed method yields a performance improvement above the naive combination and simple two-stage method.

**Questions:**

My main concern is the novelty of the proposed method. It would be good if the author can further explain on this.

**Limitations:**

No  negative societal impact is found.

**Strengths And Weaknesses:**

Strengths:
1. The writing is good and easy to follow
2. Thorough theoretical analysis is provided
3. The target problem has never been explored before
4. Unique challenge of this problem is identified and solved


Weakness:
1. Limited novelty in the proposed method. The proposed Pseudo-Label Attack is very similar to the simple two-stage baseline. Also, the warm-up attack has also been explored in the previous work [1]. Meanwhile, the performance improvement above the simple two-stage baseline is not significant.

[1] Liu, Chen, et al. "On the loss landscape of adversarial training: Identifying challenges and how to overcome them." Advances in Neural Information Processing Systems 33 (2020): 21476-21487.

---

> ### Author Response · Authors · 2022-08-01
> **Response to Reviewer u259**
>
> Thank you for reviewing our paper, with constructive comments and strong support. Here are our detailed responses to your suggestions/questions.
>
> **Q1: About the novelty of the proposed method. The performance improvement above the simple two-stage baseline is not significant.**
>
> **A1:** First, the proposed method is conceptually novel since it is naturally based on the unique challenges identified in the new setting, i.e., AT with CLs, which have both theoretical and empirical insights for the new research problem. It has a different motivation and underlying principle from other literature. Different from engineeringly simplifying the problem in the two-stage baseline, our Pseudo-label Attack (PLA) further utilizes the dynamics of adversarial optimization with CLs. Different from escaping from the suboptimal random initialization in [1], our Warm-up Attack (Warm-up) is for easing the adversarial optimization with CLs considering the CLs information in different epsilon balls.
>
> Second, except for the high-level differences, our proposed method has technical differences from the above-mentioned methods. As for PLA, the two-stage baseline does not consider the information of CLs (at the second stage), while our method adaptively considers both PLs and CLs during the optimization process. As for Warm-up, we also consider different strategies (refer to Section 4.3 and Appendix D.7) that can achieve our unique objective instead of only controlling the radius of the epsilon ball that is closely related to the research problem in [1].
>
> More importantly, our proposed method is a *unified framework* that integrates the two critical components together in a quite natural way and controls them adaptively throughout the adversarial optimization with CLs. Both two components are indispensable, and are proposed to solve the corresponding challenges of AT with CLs, based on our theoretical and empirical analysis. We kindly refer to our analysis in Section 5.2 and Appendix D.4, without either of the two components and reasonable scheduling of them, the adversarial optimization with CLs will be extremely unstable and result in a model with poor robustness, or even failure.
>
> As for the performance improvement above the two-stage baseline, as shown in Table 1, on Kuzushiji/CIFAR-10, our method could improve 4.18%/1.39% in terms of AA and 1.85%/0.90% in terms of natural accuracy, which are considered as significant improvements in the literature of adversarial training.
>
> [1] On the loss landscape of adversarial training: Identifying challenges and how to overcome them.

---

> > ### Comment · Reviewer_u259 · 2022-08-05
> > **Raise the score**
> >
> > The authors have addressed my concerns. I would raise my score.

---

### Official Review · Reviewer_gLqj · 2022-07-11

**Rating:** 6
**Confidence:** 3
**Soundness:** 2 fair
**Presentation:** 2 fair
**Contribution:** 2 fair

**Summary:**

This paper focuses on how to make adversarial training(AT) applicable in a new setting where complementary labels (CL) instead of ground-truth labels are given for AT. The authors claim that the main obstacles for CL-based AT are intractable adversarial optimization and low-quality adversarial examples. Based on this, the authors propose to solve the problems with warm-up attack and pseudo-label attack. Experiment results show that  the proposed method successfully build robust models in CL setting while many baselines fail to get a robust model.



**Questions:**

I am still not understand why it is necessary to introduce CL into AT to increase the difficulty of AT. As discussed above, why a nosied dataset for AT is not considered? I would be really appreciate if the authors can explain more why use CL to make the supervision imperfect. What is the advantage of robust model trained from CL-based AT since it performs worse than models trained with normal AT?

**Limitations:**

The authors have addressed the limitations and potential negative societal impacts.

**Strengths And Weaknesses:**

Strength: the proposed method is neat and reasonable. The authors first analyze the reasons why AT fails in CL setting and then design corresponding solutions to mitigate the problems.

Weakness:

1. I am not quite agree with the setting proposed by the authors. The AT with complementary labels considers if there is no perfect supervision data for training a robust model. However, the authors did not make it clear and reasonable that why we should consider AT in such a setting. I admit that exploring the performance of AT in noised data might be necessary and practical, but exploring the performance of AT in CL setting seems to make it unnecessarily harder for adversarial training. It can be more reasonable if the authors can demonstrate their CL-based AT performs better than vanilla AT when the training data is noised, instead of first using CL to make the supervision imperfect and then try to conduct AT under imperfect supervision.
2. Consequently, the robustness trained under CL setting does no lead to better robustness mostly because of the imperfect supervision as shown in the experiments. For most cases vanilla AT under standard setting outperforms the proposed methods in CL setting in Table 1. Though such unfair setting provides vanilla AT advantage, I still think the proposed setting is not reasonable. To better demonstrate the effectiveness, a noised dataset for AT may be considered and the authors can compare the performance on such a dataset between vanilla AT and their CL-based AT method.

---

> ### Author Response · Authors · 2022-08-01
> **Response to Reviewer gLqj: Part 1**
>
> Thanks for reviewing our paper, with the concerns about our setting. Here are our detailed responses to your suggestions/questions.
>
> **Q1: Why should we consider AT with CLs?**
>
> **A1:** First of all, ordinary training with complementary labels (i.e., complementary learning) is a promising problem setting in weakly supervised learning. It has been included as a chapter in the book "Machine Learning from Weak Supervision: An Empirical Risk Minimization Approach", Adaptive Computation and Machine Learning series, The MIT Press. Its success illustrates a possibility of training ordinary classifiers even when all the labels given for training are wrong (thus the only possibility where we cannot train classifiers is that the instances and the labels are statistically independent, i.e., the labels are completely random). Ordinary training with complementary labels can be employed in certain cases to reduce labeling burden or even make labeling a specific unlabeled dataset from impossible to possible, since the annotators do not need to be domain experts now.
>
> On the other hand, adversarial training with ordinary labels is extremely popular nowadays. According to the Test of Time Award speech of ICML 2022, there are already more than 10K related papers in the more general adversarial learning where adversarial training is a major branch. Most trustworthy machine learning and computer security researchers, if not all of them, think that adversarial robustness is already a sanity check before deploying a trained model in the real world. In other words, a model with high natural accuracy but low adversarial accuracy can only be deployed in some strictly controlled environments, while a model to be deployed in the wild must have both high natural accuracy and high adversarial accuracy.
>
> Therefore, we are interested in going one step further for ordinary training with complementary labels by taking robustness benefits from adversarial training with ordinary labels, namely, we are considering adversarial training with complementary labels. Unfortunately, naive combinations of the two cannot work under the new problem setting, and we have proposed conceptually and technically novel methods to handle the new problem setting. That being said, the new problem setting itself is of scientific interests to both research areas of weakly supervised learning and adversarial learning. Note that NeurIPS is a scientific conference, and hence usefulness is not the exclusive measure of importance, especially for papers proposing new problem settings. Only if a new problem setting has been proposed and the corresponding paper has been published, can follow-up methods be proposed, making a topic more and more popular and at the same time more and more useful in practice. We hope our effort can make adversarial training with complementary labels practically useful and let its trained models be able to be safely deployed in the real world in the near future. Please focus on science and evaluate our scientific contributions.

---

> > ### Comment · Reviewer_gLqj · 2022-08-06
> > **Response to the rebuttal**
> >
> > I appreciate the authors' detailed response, which addressed my concerns on the settings of this paper. Based on the thorough theoretical analysis as well as detailed empirical evaluation in the paper, I would like to raise my score and recommend this paper to be accepted.

---

> ### Author Response · Authors · 2022-08-01
> **Response to Reviewer gLqj: Part 2**
>
> **Q2: Other discussions about AT with imperfect supervision**
>
> **A2:**
>
> > It can be more reasonable if the authors can demonstrate their CL-based AT performs better than vanilla AT when the training data is noised, instead of first using CL to make the supervision imperfect and then try to conduct AT under imperfect supervision.
>
> First, considering AT with noisy labels and complementary labels are two different research problems. Recently, some work studied the interaction of adversarial training with noisy labels [1] (e.g., discovered the smoothing effect of AT). However, they did not target for AT under imperfect supervision, and did not mainly study how to conduct AT with noisy labels (they used AT to help label noise instead). Therefore, "*how to achieve high robustness if the clean dataset is partly corrupted*" is still an open research problem. While the CL is a special case of the NL since the clean dataset is fully corrupted. We kindly refer to our discussion in the response to Q1 of Reviewer Jbnu, we briefly design a new three-stage algorithm that leverages their empirical insights [1], and observe that its performance is not superior to the two-stage baseline, even in an unfair setting. Second, we would like to clarify that we are considering AT under CLs, instead of using CLs to further enhance AT in the normal setting with perfect supervision. Finally, we conduct the performance evaluation of vanilla AT and our method on Kuzushiji and CIFAR10 with symmetry noise rates of 95% and 100% (similar to CLs where the labels are **fully corrupted**), and show the results in Table 1. Note that we keep the same setups as described in our paper, and do not fine-tune any hyperparameters. It is clearly observed that our method performs consistently better than the vanilla AT in such cases.
>
> **Table 1.** Performance evaluation of vanilla AT and our method on the noisy dataset [Last/Best checkpoints].
>
> |                      |      Natural      |        PGD        |        CW         |        AA         |
> | :------------------: | :---------------: | :---------------: | :---------------: | :---------------: |
> | ***Kuzushiji-95%***  |                   |                   |                   |                   |
> |      Vanilla AT      |   10.00 / 10.00   |   10.00 / 10.00   |   10.00 / 10.00   |   9.97 / 10.00    |
> |         Ours         | **74.09 / 74.22** | **68.07 / 68.78** | **66.88 / 67.67** | **56.04 / 57.21** |
> | ***Kuzushiji-100%*** |                   |                   |                   |                   |
> |      Vanilla AT      |   10.00 / 10.00   |   10.00 / 10.00   |   10.00 / 10.00   |   10.00 / 10.00   |
> |         Ours         | **90.65 / 90.92** | **84.31 / 85.27** | **82.14 / 83.27** | **66.85 / 67.80** |
> |  ***CIFAR10-95%***   |                   |                   |                   |                   |
> |      Vanilla AT      |   10.00 / 10.00   |   10.00 / 10.00   |   10.00 / 10.00   |   10.00 / 10.00   |
> |         Ours         | **26.92 / 27.34** | **21.92 / 22.10** | **21.67 / 21.79** | **21.59 / 21.71** |
> |  ***CIFAR10-100%***  |                   |                   |                   |                   |
> |      Vanilla AT      |   10.00 / 10.00   |   10.00 / 10.00   |   10.00 / 10.00   |   10.00 / 10.00   |
> |         Ours         | **64.02 / 63.84** | **41.98 / 42.33** | **40.04 / 40.47** | **39.23 / 39.75** |
>
> [1] Understanding the Interaction of Adversarial Training with Noisy Labels.

---

> ### Author Response · Authors · 2022-08-02
> **Response to Reviewer gLqj: Part 3**
>
> **Q2: Other discussions about AT with imperfect supervision**
>
> **A2:**
>
> > The robustness trained under CL setting does not lead to better robustness mostly because of the imperfect supervision. For most cases vanilla AT under standard setting outperforms the proposed methods in CL setting in Table 1; What is the advantage of robust model trained from CL-based AT since it performs worse than models trained with normal AT?"
>
> The performance of AT with ordinary labels is **served as an oracle, and a reference** (measuring current performance gaps between AT with CLs and AT with ordinary labels) for future research in the studied problem setting,  i.e., AT with CLs.  Hence, it is not comparable to the methods with different settings. On the other hand, the vanilla AT would fail given complementary labels, and the naive combinations of vanilla AT with complementary learning methods have been widely verified not to be better than our proposed method.
>
> > I still not understand why it is necessary to introduce CL into AT to increase the difficulty of AT; why use CL to make the supervision imperfect.
>
> We would like to mention that we are not introducing CLs deliberately to AT to achieve some objectives (e.g., try to lead to better robustness).  In this paper, what we are focusing on is **"how to equip the machine learning model with adversarial robustness when we only have complementary labels in the dataset"**, which is a scientific/research problem, not an engineering one. We kindly refer to our response to Q1 for the motivation and research significance of AT with CLs.
>
> Due to the page limit, part of the discussion about the motivation and significance of AT with CLs is added to Appendix E, and will be moved to the introduction later.

---

### Official Review · Reviewer_GzvY · 2022-07-11

**Rating:** 8
**Confidence:** 4
**Soundness:** 3 good
**Presentation:** 3 good
**Contribution:** 3 good

**Summary:**

The authors explore a brand new yet challenging setting that studies adversarial training (AT) with complementary labels (CL). Generally, it is significant to involve CL in AT since imperfect supervision can be common in real AT scenarios. However, the direct combination of AT and CL consistently leads to failure according to extensive empirical observations.

To explore this issue, the authors provide theoretical evidence that there exists inconsistency between complementary risk and ordinary risk of adversarial optimization with limited CLs. Together with the empirical studies of gradients, they identify two key challenges including intractable adversarial optimization and low-quality adversarial examples.

Based on the analysis, a new attack strategy is introduced. A warm-up is adopted to ease the difficulty of adversarial optimization. With the model prediction as supplementary information, the adversarial training gradually involves the pseudo label predicted by the model. The authors conduct extensive experiments on different datasets and compare proposed algorithm with various baselines to demonstrate its effectiveness.

**Questions:**

1.Please provide empirical evaluation of pseudo labels.
2.Please include more ablation studies of warm-up strategies of attacks.

**Limitations:**

Yes, the authors have discussed the limitations and potential negative societal impacts in Appendix E.

**Strengths And Weaknesses:**

Pros:

* In general, this paper is well-written and easy to follow. The motivation is clear. The studied setting is both significant and challenging.
* Both the theoretical analysis of the inconsistency between empirical risks with the assumption of limited CLs and the empirical analysis of the difficulty of adversarial optimization as well as adversarial example generation are intriguing.
* The proposed techniques including warm-up and pseudo-label are proposed based on the theoretical and empirical analysis, which are simple yet natural.
* The authors provide sufficient evaluation of proposed algorithm. The evaluation is conducted on MNIST, Kuzushiji, CIFAR-10 and SVHN, and includes various SOTA complementary losses for comparison. The proposed algorithm consistently achieves better adversarial robustness as well as stability.

Cons:

* In Section 4.3, the authors propose to use model prediction as a strong supplementary information since the model tends to assign high confidence to ordinary label when the epsilon ball is small enough. However, it is difficult to find empirical evidence of it in experimental section. It would be better for the authors to report the accuracy of pseudo labels in some scenarios.
* The author introduces a warm-up attack which controls the radius of epsilon ball. However, the number of attack steps is fixed during warm-up. It would be better for the authors to conduct more ablation studies of it.

---

> ### Author Response · Authors · 2022-08-01
> **Response to Reviewer GzvY: Part 1**
>
> Thank you for reviewing our paper, with constructive comments and strong support. Here are our detailed responses to your suggestions/questions.
>
> **Q1: Empirical evaluation of pseudo-labels.**
>
> **A1:** Thanks for pointing this out, we conduct two experiments for the empirical evaluation of pseudo-labels to comprehensively verify this point.
>
> First, we report the Acc. of pseudo-labels (%) w.r.t. the slowly increased $\epsilon$ ball in Table 1, where the $\epsilon$ ball is relatively small. To be specific, the radius of the epsilon ball is increased from 0 to 0.3 within $E_\mathrm{s}=50$ epochs, following our proposed scheduler as shown in Section 4.3 and Figure 3(a). To avoid the effect of (natural complementary learning) warmup period on the analysis of accuracy dynamics of pseudo-labels, we set the epoch of warmup $E_\mathrm{i}=0$. We keep the other setups fixed and rerun the experiments on Kuzushiji dataset.
>
> The results show that the accuracy of pseudo-labels rises up rapidly at the early stage of adversarial optimization with CLs, which demonstrates that a small epsilon ball (e.g., $\epsilon$ is increased from 0 to 0.029 within the first 10 epochs) is helpful for the formation of a discriminative model that tends to assign high confidence to the ordinary labels.
>
> **Table 1.** Acc. of pseudo-labels (%) w.r.t. the slowly increased $\epsilon$ ball in each Epoch.
>
> |            Epoch             |        1         |        2         |        3         |        4         |        5         |        6         |        7         |        8         |        9         |        10        |        50        |
> | :--------------------------: | :--------------: | :--------------: | :--------------: | :--------------: | :--------------: | :--------------: | :--------------: | :--------------: | :--------------: | :--------------: | :--------------: |
> | **Acc of pseudo-labels (%)** | 12.52($\pm$0.42) | 25.44($\pm$2.26) | 46.78($\pm$4.27) | 61.57($\pm$4.53) | 71.81($\pm$6.13) | 79.70($\pm$4.20) | 85.27($\pm$1.63) | 87.55($\pm$1.05) | 89.00($\pm$1.05) | 90.39($\pm$1.15) | 97.22($\pm$0.18) |
> |         $\epsilon_e$         |      0.0003      |      0.0012      |      0.0027      |      0.0047      |      0.0073      |      0.0105      |      0.0143      |      0.0186      |      0.0234      |      0.0286      |      0.3000      |
> | $\epsilon_e/\epsilon_{\max}$ |      0.10%       |      0.39%       |      0.89%       |      1.57%       |      2.45%       |      3.51%       |      4.76%       |      6.18%       |      7.78%       |      9.55%       |     100.00%      |
>
> Second, we report the Acc. of pseudo-labels (%) w.r.t. the rapidly increased $\epsilon$ ball in Table 2, where we only modifying $E_\mathrm{s}=10$ without changing other setups.
>
> In this way, at the beginning of training, the adversarial data are found within a relatively big epsilon ball (also with a rapid growth rate). The results demonstrate that the model fails to assign high confidence to the ordinary label in such a case, and even may fail the adversarial optimization (observed in 2 out of 3 runs).
>
> **Table 2.** Acc. of pseudo-labels (%) w.r.t. the rapidly increased $\epsilon$ ball in each Epoch.
>
> |            Epoch             |        1         |        2         |        3         |        4         |        5        |        6         |        7         |        8         |        9         |        10        | 50               |
> | :--------------------------: | :--------------: | :--------------: | :--------------: | :--------------: | :-------------: | :--------------: | :--------------: | :--------------: | :--------------: | :--------------: | ---------------- |
> | **Acc of pseudo-labels (%)** | 11.08($\pm$0.04) | 12.66($\pm$1.21) | 12.93($\pm$1.48) | 12.98($\pm$1.52) | 13.0($\pm$1.55) | 13.02($\pm$1.56) | 13.02($\pm$1.56) | 13.02($\pm$1.56) | 13.02($\pm$1.56) | 13.01($\pm$1.55) | 13.02($\pm$1.56) |
> |         $\epsilon_e$         |      0.0073      |      0.0286      |      0.0618      |      0.1036      |     0.1500      |      0.1964      |      0.2382      |      0.2714      |      0.2927      |      0.3000      | 0.3000           |
> | $\epsilon_e/\epsilon_{\max}$ |      2.45%       |      9.55%       |      20.61%      |      34.55%      |     50.00%      |      65.45%      |      79.39%      |      90.45%      |      97.55%      |     100.00%      | 100.00%          |
>
> Overall, the results demonstrate the model tends to assign high confidence to ordinary labels when the epsilon ball is small. Along with training, we can obtain the pseudo-labels with high accuracy using our proposed cashed (exponential moving average) probability. We update the whole results in Appendix D.6 and Figure 9 (a).

---

> ### Author Response · Authors · 2022-08-01
> **Response to Reviewer GzvY: Part 2**
>
> **Q2: Ablation study on strategies of Warm-up attack.**
>
> **A2:** Thanks for your suggestion. We try two more strategies of Warm-up attack, and conduct experiments on Kuzushiji dataset: (a) we only change the number of attack steps $k$; (b) similar to the original implementation, we still control the radius of the epsilon ball, but accompanied by the proportional increase of $k$ instead of the step size $\alpha$. We summarize the results in Table 3.
>
> **Table 3.** Ablation study on strategies of Warm-up Attack [Last/Best checkpoints].
>
> |          |               Natural                |                 PGD                  |                  CW                  |                  AA                  |
> | :------: | :----------------------------------: | :----------------------------------: | :----------------------------------: | :----------------------------------: |
> | original | 91.26($\pm$0.45) / 91.60($\pm$0.49)  | 84.96($\pm$0.46) / 85.88($\pm$0.48)  | 82.79($\pm$0.50) / 83.74($\pm$0.35)  | 67.51($\pm$0.74) / 68.75($\pm$0.68)  |
> |   (a)    | 60.09($\pm$35.42) / 89.22($\pm$0.66) | 55.55($\pm$32.21) / 81.94($\pm$0.90) | 54.02($\pm$31.13) / 79.14($\pm$0.93) | 45.79($\pm$25.31) / 64.41($\pm$0.94) |
> |   (b)    | 91.48($\pm$0.39) / 91.82($\pm$0.39)  | 85.43($\pm$0.23) / 86.12($\pm$0.37)  | 83.26($\pm$0.10) / 84.04($\pm$0.43)  | 67.52($\pm$1.37) / 69.43($\pm$0.40)  |
>
> The results demonstrate that trial (a) may suffer from unstable adversarial optimization, while trial (b) is comparable and slightly better than our original implementation. We update the detailed results with corresponding analysis in Appendix D.7 and Figure 9(b).

---

### Author Response · Authors · 2022-08-01
**General Response to All Reviewers**

We sincerely appreciate all reviewers' time and efforts in reviewing our paper as well as the comments. We have updated our draft, including the additional experiments required by each reviewer, and the motivation and scientific significance of our studied setting.

In addition to the pointwise responses below, here we summarize our updates.

- [Setting] We explain and highlight the motivation and scientific significance of our studied new setting (in Appendix E), namely, adversarial training with complementary labels (AT with CLs). The new problem setting itself is of scientific interests to both research areas of weakly supervised learning and adversarial learning. The studied setting has never been explored, and we hope our effort can make AT with CLs practically useful and let its trained models be able to be safely deployed in the real world in the near future.

- [Method Novelty] We compare and highlight the novelty of our proposed method (in Appendix D.4). We proposed a unified framework, which naturally combines and adaptively controls two indispensable components (Warm-up Attack and Pseudo-labels Attack) throughout the adversarial optimization with CLs. Conceptually, the motivation and underlying principle of the two components in our method are closely related to the unique challenges identified in our new problem setting, which have never been revealed. Technically, the two components utilize the dynamic information of complementary labels during the adversarial optimization, which is also not been considered in previous literature. Comprehensive experiments are conducted to verify its rationality and effectiveness.
- [Extra Experiments] We add the experiments about the empirical evaluation of pseudo-labels attacks (in Appendix D.6), various strategies of warm-up attack (in Appendix D.7), comparison with our newly designed methods for AT under noisy labels, and verification of proposed methods combined with more complementary learning baselines (in Appendix D.4).

We hope our responses below could address the reviewers' concerns, and we are welcome to discuss further if any point is unclear.

The authors of Paper1958.

---

### Author Response · Authors · 2022-08-02
**Further clarification about the problem settings of noisy and complementary labels**

Since there is a shared concern about the two problem settings *learning with noisy labels* and *learning with complementary labels*, we'd like to further clarify their connections and differences with some details.

A quick conclusive message is that although CL as a problem setting is a special case of NL, **not every algorithm designed for the NL problem can be employed to solve the CL problem**. When talking about problem settings, supervised learning (where all labels are correct) and learning with CLs (where all labels are incorrect) are **the two ends of a line** named learning with NLs. Note that the range of this line is too wide, and we cannot require or expect an algorithm to cover the full range. The *critical point* is when all labels are random in which case learning a classifier is indeed impossible. For any algorithm working on the CL side of the NL line, a special knowledge (i.e., the problem is on the CL side) is needed in the algorithm design. This makes sense --- if we simply apply a general-purpose NL algorithm to supervised learning, it doesn't work well either (due to unnecessary robustness reducing statistical efficiency); similarly, if we simply apply a general-purpose NL algorithm to the CL problem, unfortunately, it might not work at all.

Some machine learning methods such as *robust loss* and *robust regularization* are designed for robustness in the general sense rather than robustness against NLs. They can only work in a narrow range of the NL line when the noise rate is low enough. *Loss correction*, *sample selection*, and *label correction* are specially designed for robustness against NLs and thus can work in a wide range of the NL line when the noise rate is fairly high, provided that **correct labels can still dominate incorrect labels**. On class-balanced benchmarks (such as the MNIST family and CIFAR-10), we theoretically need $T$ to be diagonally dominant to enable *learnability* in the asymptotic case, and we empirically need $T$ to be **diagonally dominant by a sufficient margin** to enable *stable training* in the finite-sample case.

- More specifically, sample selection and label correction require the row diagonal dominance of $T$ --- for example, on MNIST and CIFAR-10, the noise rate should be lower than 81% for symmetric noise (where the original class gets 19% of its data and each of other classes gets 9% data of that class) or 45% for pairwise noise (where the original class gets 55% of its data and the next class gets 45% data of that class), so we require a margin of 10% beyond diagonal dominance.

- On the other hand, loss correction itself has no constraint on $T$ if it is known in advance, while the estimation of $T$ requires its column diagonal dominance, or equivalently, $\arg\max_{\tilde{y}}p(\tilde{y}|x)==\arg\max_{y}p(y|x)$ for all $x$, to determine the class membership of given or found (likely) *anchor points*.

Therefore, we can see that only backward/forward loss corrections and related methods on top of them can work for the problem of learning with CLs, and at the same time the derived algorithms must also utilize the special knowledge about $T$.

Regarding the **unpublished** paper entitled "Understanding the Interaction of Adversarial Training with Noisy Labels" (here, the word "with" modifies interaction rather than training), note that the goal is indeed not about adversarial training with noisy labels, but about **using the number of PGD steps to improve sample selection quality and thus improve the natural accuracy of learning with NLs**. According to the one-sentence summary of that paper,

> Adversarial training (AT) itself is noisy labels (NL) correction; "PGD step number" in AT is a new criterion for sample selection.

and according to its abstract,

> Firstly, we find if a point is too close to its noisy-class boundary (e.g., one step is enough to attack it), this point is likely to be mislabeled, which suggests to adopt the number of PGD steps as a new criterion for sample selection to correct NL. Secondly, we confirm that AT with strong smoothing effects suffers less from NL (without NL corrections) than standard training, which suggests that AT itself is an NL correction.

As a result, it is even not possible to directly apply that method in a two-stage manner (i.e., sample selection --> adversarial training). On the CL side of the NL line, none labels are correct and thus **there is no sample to select for standard or adversarial training**. Therefore, even if the aforementioned paper has been published, it would not compromise the novelty of the current paper under consideration.

Last but not least, note that complementary labels are the most adversarial (yet structured) label modification, whereas adversarial examples are the most adversarial instance modification. As a consequence, showing the possibility of better classifier training than first CL and then AT is really not a small step and needs a lot of unique insights.

---

### Meta-Review · Area_Chair_DXjR · 2022-08-26

**Recommendation:** Accept
**Confidence:** Certain

**Metareview:**

This paper focuses on a significant and challenging problem: adversarial training (AT)  with complementary labels. A naive combination of AT with existing complementary learning techniques fails to achieve good performance. The authors conduct both theoretical and empirical analyses of this phenomenon and identified two key challenges including intractable adversarial optimization and low-quality adversarial examples. Furthermore, two attack approaches are proposed accordingly: a warm-up attack to ease the adversarial optimization and a pseudo-label attack to improve the adversarial example quality. All reviewers recognize the effectiveness of the proposed method through experimental evaluations.  During the discussion, the authors also successfully addressed the reviewers' questions on the problem settings, the novelty of the pseudo-label attack, warm-up strategies, etc.  Based on the positive reviews and thorough discussions, we recommend the acceptance of the paper.


**Award:**

No

---

### Decision · Program_Chairs · 2022-09-14

Accept